# Beyond Magnitude: Scale-Invariant Evidential Fusion for Multi-View Classification

**Wei Liu**[1]  **Yufei Chen**[1 †]  **Jie Shi**[2]  **Xiaodong Yue**[3]

## Abstract

Evidential Deep Learning (EDL) enables trustworthy multi-view classification, yet suffers from a critical vulnerability: the *Scale Mismatch Problem*. We theoretically demonstrate that existing evidential fusion rules erroneously equate logit magnitude with semantic confidence, rendering them susceptible to "semantic hijacking" by inflated but uninformative views. To resolve this, we propose *Scale-Invariant Evidential Fusion (SAEF)*, a framework utilizing instance-wise standardization to strictly decouple confidence from scale. Instead of relying on magnitude dominance, SAEF aggregates views based on statistical consensus. Theoretically, SAEF guarantees invariance to global scaling and robustness to asymmetric dominance. Experiments on four diverse datasets confirm that SAEF outperforms state-of-the-art baselines in accuracy and robustness to semantic conflicts and noise, ensuring stability against severe scale perturbations.

## 1. Introduction

Multi-view learning aims to integrate heterogeneous information for robust decision-making (Guo et al., 2024; Yang et al., 2025b; Fu et al., 2024; Yang et al., 2025a; Liu et al., 2025a; Yang et al., 2026). However, standard deterministic models often function as black boxes, inadvertently overtrusting noisy or conflicting views in safety-critical scenarios. To quantify reliability, Evidential Multi-view Classification (EMC) (Han et al., 2022; Liu et al., 2024; Liang et al., 2025b; Liu et al., 2025d) has been developed, building upon Evidential Deep Learning (EDL) (Sensoy et al., 2018). By modeling class probabilities as Dirichlet distributions, EMC

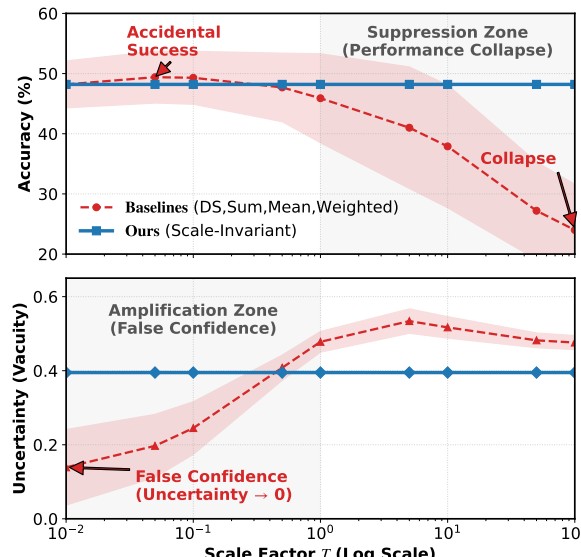

*Figure 1.* **Scale Sensitivity Analysis**. We scale the logits of a single view by $1/T$ across a wide range of $T \in [0.01, 100]$ on SUNRGBD. **Standard methods (Red dashed line)**, representative of four fusion rules which exhibit negligible differences ($< 0.1\%$), are evaluated against our approach. **(Top)** Baselines achieve peak accuracy at $T < 1$ merely through "Accidental Success", while suffering collapse at $T > 1$. **(Bottom)** The danger of amplification is revealed by the vanishing evidential uncertainty ($\to 0$), indicating False Confidence. In contrast, **Ours (Blue)** maintains consistent performance and reliability (Scale Invariance). Red shaded regions denote the standard deviation across 5 folds.

aggregates views at the evidence level, offering a probabilistic framework to estimate uncertainty based on *evidence vacuity* rather than mere probability scores. This enables the system to explicitly identify the lack of evidence (ignorance) distinct from predictive ambiguity, providing a mechanism to dynamically down-weight uninformative views.

In the standard EMC paradigm, evidence is extracted from backbones and aggregated using canonical evidence fusion rules, such as *Dempster-Shafer (DS) fusion* (Han et al., 2022), *Sum fusion* (Liu et al., 2022), *Mean fusion* (Liang et al., 2025b), or *Weighted fusion* (Xu et al., 2025). While effective under ideal conditions, these strategies rely on a strong implicit assumption: that the *magnitude of evidence* (directly determined by the logit scale) is consistently

[1]School of Computer Science and Technology, Tongji University, Shanghai, China. [2]School of Computer Engineering and Science, Shanghai University, Shanghai, China. [3]Institute of Artificial Intelligence, Shanghai University, Shanghai, China. Correspondence to: Yufei Chen <yufeichen@tongji.edu.cn>.

*Proceedings of the 43$^{rd}$ International Conference on Machine Learning*, Seoul, South Korea. PMLR 306, 2026. Copyright 2026 by the author(s).

aligned across views and strictly proportional to the view's semantic reliability.

However, this assumption of *aligned scales* is rarely satisfied in real-world environments. We identify a prevalent **Scale Mismatch Problem**, where evidential magnitudes drift arbitrarily due to uncontrolled factors such as *sensor malfunctions* (e.g., signal amplification/attenuation), *domain shifts*, or *corruptions* (Hendrycks & Dietterich, 2019; Wei et al., 2022). Crucially, standard fusion rules are logically sensitive to such global scale shifts. Since evidential magnitude is conflated with reliability, an arbitrarily inflated noisy or conflicting view can hijack the fusion process via *false dominance* (False Confidence), while a suppressed reliable view is misinterpreted as ignorance and discarded (*signal suppression*). This vulnerability renders the fusion process unstable against scale perturbations, as confirmed by our motivational analysis.

To empirically dissect this vulnerability, we simulate these scale perturbations in Figure 1 by applying temperature scaling ($\mathbf{z}/T$) to the logits $\mathbf{z}$ of a single view on SUNRGBD (while keeping others fixed). The results expose a fundamental dilemma between performance and safety in standard fusion strategies: **1) The Safety Trap (Amplification Zone, $T < 1$):** Simulating signal amplification, baselines appear to achieve peak accuracy (labeled as "Accidental Success"). However, the bottom plot reveals the hidden danger: uncertainty collapses significantly ($\approx 0.14$). This confirms that fusion outcome is mathematically dominated by the sheer magnitude of the amplified view. Crucially, this success is merely "accidental", as it hinges on the premise that the dominant view happens to be correct. Had this amplified view been adversarial or corrupted, the model would indiscriminately output a high-confidence error. Thus, while accuracy remains high on clean data, the accompanying *False Confidence* renders the system incapable of distinguishing valid evidence from numerical inflation. **2) Performance Collapse (Suppression Zone, $T > 1$):** Conversely, representing signal attenuation, informative signals are drowned out, causing accuracy to plummet catastrophically (e.g., from 48% to 24%). These observations confirm that existing methods fundamentally conflate *numerical magnitude* with *semantic evidence*, forcing a trade-off where one must choose between being "unsafe but accurate" (amplification) or "safe but useless" (suppression).

This flaw is particularly pernicious in safety-critical scenarios: For instance, in conflicting situations (e.g., sensor hijacking or obstruction), a corrupted view often manifests with abnormally high magnitudes. Driven by scale sensitivity, standard methods will inadvertently capitulate to this "loudest" but incorrect view, resulting in high-confidence errors (i.e., failure to trigger uncertainty alerts). **Therefore, robust fusion requires a mechanism that is logically in-variant to such magnitude shifts, trusting only the structural consensus of evidence rather than its absolute scale.**

To achieve this, we propose Scale-Invariant Evidential Fusion (SAEF), a framework that explicitly **decouples reliability from evidential magnitude**. Diverging from canonical approaches that passively accept raw logit scales, SAEF introduces *Instance-wise Evidential Standardization* to anchor view reliability on the statistical significance of the class distribution rather than absolute numerical values. These standardized beliefs are then robustly synthesized via *Uncertainty-Weighted Stouffer Fusion*. As shown in Figure 1, SAEF neutralizes scale bias, maintaining robust performance across orders of magnitude of perturbation without requiring additional learnable parameters. Our main contributions are summarized as follows:

- We identify the *Scale Mismatch Problem* and theoretically prove that canonical fusion strategies (e.g., DS, Sum, Mean, Weighted) inherently conflate logit magnitude with semantic reliability, rendering them pathologically sensitive to uninformative scale variations.

- We propose *Scale-Invariant Evidential Fusion (SAEF)*, a statistically grounded framework with zero additional learnable parameters. By integrating instance-wise standardization with Stouffer's fusion, we provide rigorous theoretical guarantees for *Global Scale Invariance* and robustness against asymmetric dominance.

- Extensive experiments on diverse benchmarks demonstrate that SAEF significantly outperforms state-of-the-art methods. Crucially, SAEF exhibits superior stability against severe *scale perturbations*, *sensor noise*, and *multi-view conflicts*, validating its resilience in scenarios where baselines suffer from catastrophic collapse.

## 2. Related Work

**Evidential Multi-view Classification:** Evidential Deep Learning (EDL) (Sensoy et al., 2018) formulates classification as an evidence collection process parameterized by Dirichlet distributions, explicitly capturing *evidence vacuity* as evidential uncertainty. Consequently, EDL has been successfully adapted across numerous domains, including fine-grained classification (Xu et al., 2023), medical data analysis (Fu et al., 2023; Li et al., 2025), hallucination detection (Shi et al., 2026), and object detection (Xia et al., 2026). Such explicit uncertainty modeling provides a theoretical basis for quantifying view-specific reliability, enabling the system to explicitly identify and down-weight unreliable sources (e.g., noisy or out-of-distribution views) during fusion. Building on this, EMC leverages Dempster-Shafer Theory (DST) (Fu et al., 2022; Lv et al.,

2021) and Subjective Logic (SL) (Liu et al., 2023) to aggregate view-specific evidence. Canonical fusion strategies primarily include *DS fusion* (Han et al., 2022; Liang et al., 2025a) (orthogonal sum), *Sum fusion* (Liu et al., 2022; Zhang et al., 2022) (linear accumulation), *Mean fusion* (Xu et al., 2024; Liang et al., 2025b) (belief averaging), and *Weighted fusion* (Xu et al., 2025) (dynamic re-weighting). These methods aim to synthesize beliefs while maintaining robustness against unreliable or conflicting evidence.

**Limitations of Prior Art:** Despite these advancements, a critical oversight persists across these strategies: *the assumption of scale alignment*. They operate directly on raw evidence magnitudes, implicitly assuming that logit scale is proportional to reliability. As we identify, this renders EMC methods vulnerable to the *Scale Mismatch Problem*, where reliable low-magnitude views are suppressed and noisy high-magnitude views hijack the decision, which is a limitation our work explicitly addresses.

## 3. Revisiting Evidential Multi-view Fusion

### 3.1. Formulation of View-specific Evidence

Consider a multi-view classification task with $V$ views $\mathcal{X} = \{\boldsymbol{x}^v\}_{v=1}^V$ and label $y \in \{1, \dots, K\}$. We adopt EDL to model class probabilities via Dirichlet distributions, enabling the explicit quantification of *evidence vacuity* as a measure of reliability for robust fusion.

For view $v$, let $f^v(\cdot)$ denote the backbone network that maps the input $\boldsymbol{x}^v$ to a logit vector $\boldsymbol{z}^v \in \mathbb{R}^K$. An activation function $a(\cdot)$ (e.g., Softplus) is applied to induce the non-negative *evidence vector* $\boldsymbol{e}^v = a(\boldsymbol{z}^v) = \{e_k^v\}_{k=1}^K$. This evidence parameterizes a Dirichlet distribution $\mathrm{Dir}(\boldsymbol{\alpha}^v)$ where $\boldsymbol{\alpha}^v = \boldsymbol{e}^v + \mathbf{1}$. Consequently, the total evidence strength $S^v = \sum_{k=1}^K \alpha_k^v$ determines the view's evidential uncertainty (vacuity), defined as $u^v = K/S^v$.

### 3.2. Unification of Canonical Evidential Fusion Rules

To synthesize heterogeneous information, EMC employs fusion mechanisms to integrate view-specific evidence into a unified joint representation $\boldsymbol{e}^{joint}$. While grounded in DST or Subjective Logic, recent works (Liu et al., 2025c; Lu et al., 2025) reveal that canonical fusion strategies admit efficient algebraic equivalents directly in the evidence space. Here, **we unify them as algebraic operations on evidence magnitudes** (derivations are shown in Appendix A):

**DS Fusion** (Han et al., 2022; Lu et al., 2025): Derived from Dempster's orthogonal sum. Specifically, for two views (which extends recursively):

$$\boldsymbol{e}^{joint} = \boldsymbol{e}^1 + \boldsymbol{e}^2 + \frac{1}{K}(\boldsymbol{e}^1 \odot \boldsymbol{e}^2). \tag{1}$$

**Sum Fusion** (Liu et al., 2022): Corresponds to the linear accumulation of belief consensus:

$$\boldsymbol{e}^{joint} = \sum_{v=1}^V \boldsymbol{e}^v. \tag{2}$$

**Mean Fusion** (Xu et al., 2024; Liang et al., 2025b): Computes the arithmetic average of evidence:

$$\boldsymbol{e}^{joint} = \frac{1}{V} \sum_{v=1}^V \boldsymbol{e}^v. \tag{3}$$

**Weighted Fusion** (Xu et al., 2025): A convex combination prioritizing confident views:

$$\boldsymbol{e}^{joint} = \sum_{v=1}^V \tilde{w}_v \boldsymbol{e}^v, \quad \text{where } \tilde{w}_v = \frac{1 - u^v}{\sum_{j=1}^V (1 - u^j)}. \tag{4}$$

**Remark.** While structurally diverse, these canonical strategies share a critical characteristic: **they operate directly on the absolute magnitudes of $\boldsymbol{e}^v$**. Since $\boldsymbol{e}^v$ is strictly tied to the logit scale $\boldsymbol{z}^v$, any fluctuation in logit scale directly propagates to the fused result, or even amplifies non-linearly (e.g., the quadratic term in DS fusion). This intrinsic dependency sets the stage for the *Scale Mismatch Problem* we address next.

### 3.3. The Scale Mismatch Problem

We argue that this magnitude-dependency is fundamentally flawed. In deep learning, logit magnitude is often an unreliable artifact of optimization (e.g., dependent on weight initialization or lack of regularization) rather than a true reflection of semantic confidence (Guo et al., 2017). Relying on raw magnitudes renders fusion vulnerable to "semantic hijacking", where noisy/conflicting views blindly dominate decision-making.

Formally, let $T \in \mathbb{R}^+$ be a temperature scaling factor acting on the logit vector $\boldsymbol{z}^v$. The perturbed evidence is given by $\tilde{\boldsymbol{e}}^v(T) = a(\boldsymbol{z}^v/T)$. We identify two critical failure modes (Proofs in Appendix B.1)):

**Definition 3.1 (Evidential Scale Ambiguity).** A reliable uncertainty estimator should be invariant to affine transformations that preserve the semantic direction of predictions. However, for standard EDL models, the evidential uncertainty $u^v$ is strictly coupled with the temperature $T$:

$$\lim_{T \to 0} u^v(T) = 0 \quad \text{(False Confidence)},$$
$$\lim_{T \to \infty} u^v(T) = u_{\max} \quad \text{(False Ignorance)}. \tag{5}$$

where $u_{\max}$ represents the activation-dependent maximum uncertainty capacity (e.g., $u_{\max} = 1$ for ReLU).

This ambiguity implies that a view can be arbitrarily suppressed or amplified solely by scaling its logits, independent of its informational content. When such misaligned views are aggregated, fusion strategies suffer from catastrophic instability. We categorize this failure into two regimes:

**Proposition 3.2** (**Scale Sensitivity of Fusion Strategies**). *Let $T$ be the temperature scaling factor applied to input logits. Assuming the activation function exhibits asymptotic homogeneity (i.e., $a(z/T) \approx \frac{1}{T}a(z)$ as $T \to 0$, strictly satisfied by ReLU), standard fusion strategies exhibit order-dependent sensitivity to $T$:*

*1) **Linear Sensitivity (Sum, Mean, Weighted):** These strategies propagate scale linearly. The magnitude of the fused evidence scales as $\mathcal{O}(1/T)$. Consequently, $T \to 0$ leads to linear inflation (False Confidence).*

*2) **Quadratic Sensitivity (DS):** Due to the multiplicative interaction term, DS fusion suffers from quadratic sensitivity, where the fused magnitude scales as $\mathcal{O}(1/T^2)$, causing explosive instability under global amplification.*

**Corollary 3.3** (**Asymmetric Dominance in Conflict Scenarios**). *Consider a conflict scenario where a reliable View 1 is challenged by a conflicting View 2 subjected to amplification ($T \to 0$). Canonical fusion rules fail to preserve the correct prediction. Specifically, the fused evidence aligns strictly with the amplified view:*

$$\lim_{T \to 0} \arg\max(\tilde{\boldsymbol{e}}^{joint}) = \arg\max(\boldsymbol{e}^2). \tag{6}$$

*This implies that standard paradigms are vulnerable to **false dominance**: a misleading yet high-magnitude view hijacks the decision logic, degrading fusion into a "loudest voice wins" paradigm.*

**Mechanism Analysis.** While Proposition 3.2 explains calibration failure (False Confidence), this corollary identifies the root cause of *accuracy degradation*. For linear strategies (Sum/Mean), the mechanism is straightforward: the inflated view ($1/T \gg 1$) mathematically dominates the sum. Crucially, for DS fusion, although it exhibits quadratic sensitivity under global scaling, the situation worsens in *asymmetric conflict*. Since the views are conflicting (i.e., orthogonal semantics), the quadratic interaction term $\boldsymbol{e}^1 \odot \boldsymbol{e}^2$ is effectively suppressed. Consequently, the fusion is dominated by the linear term of the amplified view (i.e., the $\frac{1}{T}\boldsymbol{e}^2$ component), similarly resulting in prediction hijacking. Thus, regardless of algebraic order, magnitude-dependency renders all canonical rules susceptible to catastrophic failure.

## 4. Method

### 4.1. Scale-Invariant Evidential Fusion (SAEF)

To address the scale sensitivity issues, we propose Scale-Invariant Evidential Fusion (SAEF). The core is to decouple

the *semantic direction* of predictions from their *magnitude*, performing fusion in a statistically standardized manifold. SAEF consists of two stages: Instance-wise Evidential Standardization and Uncertainty-Weighted Stouffer Fusion.

#### 4.1.1. INSTANCE-WISE EVIDENTIAL STANDARDIZATION

Standard fusion fails because it operates on raw logits $\boldsymbol{z}^v$ with arbitrary scales. By applying instance-wise standardization, we explicitly discard the absolute magnitude (which is sensitive to $T$) while strictly preserving the relative semantic sharpness of the class distribution. To achieve this, we project the raw logits onto a canonical evidence manifold.

For each view $v$, we compute the instance-level statistics: $\mu^v = \frac{1}{K}\sum_k z_k^v$ and $\sigma^v = \sqrt{\frac{1}{K}\sum_k (z_k^v - \mu^v)^2}$. The standardized logit $\hat{\boldsymbol{z}}^v$ is defined as:

$$\hat{\boldsymbol{z}}^v = \beta \cdot \frac{\boldsymbol{z}^v - \mu^v}{\max(\sigma^v, \epsilon)}. \tag{7}$$

This formulation imposes two critical theoretical constraints to counteract the failure modes defined in Definition 3.1:

**1) Variance Gating ($\epsilon$):** The noise floor $\epsilon$ prevents the amplification of pure noise. When a view lacks informative variation ($\sigma^v < \epsilon$), the normalization gain is bounded, preventing tiny fluctuations from being amplified; for nearly flat logits, this yields an approximately uniform distribution.

**2) Manifold Rescaling ($\beta$):** Standardization constrains logits to $\mathcal{N}(0,1)$, which inherently limits the magnitude of generated evidence (e.g., $\mathbb{E}[e_k^v] \approx 1$). In the EDL framework ($u^v = K/S^v$), this insufficient evidence strength forces the model into an *Uncertainty Plateau*, where it is mathematically incapable of expressing high confidence even for correct predictions. The canonical scale $\beta$ serves as a restoration factor. Acting as a fixed constant (unlike variable raw logits), it aligns all views to a shared semantic space by projecting standardized logits into the linear regime of the activation function (e.g., Softplus), ensuring the model recovers the capacity to express low uncertainty ($u \to 0$) for high-quality views, independent of the original feature norms.

#### 4.1.2. UNCERTAINTY-WEIGHTED STOUFFER FUSION

Simply summing standardized evidence $\hat{\boldsymbol{e}}^v = a(\hat{\boldsymbol{z}}^v)$ would discard the reliability information encoded in evidential uncertainty. We adopt a statistically grounded approach inspired by Stouffer's Z-score method (Kim et al., 2013), converting fusion into a hypothesis testing problem in the Gaussian domain.

We first map the evidential probabilities back to the standard normal space (Z-space). Let $\hat{\boldsymbol{\alpha}}^v = \hat{\boldsymbol{e}}^v + \boldsymbol{1}$ be the standardized Dirichlet parameters. The expected probability is derived as $\hat{\boldsymbol{p}}^v = \hat{\boldsymbol{\alpha}}^v/\hat{S}^v$. We compute the Z-scores using

the inverse Cumulative Distribution Function (CDF) of the standard normal distribution ($\Phi^{-1}$):

$$\boldsymbol{Z}^v = \Phi^{-1}(\text{clip}(\hat{\boldsymbol{p}}^v, \delta, 1 - \delta)), \tag{8}$$

where $\delta = 10^{-6}$ ensures numerical stability. $\boldsymbol{Z}^v$ represents the *standardized logit in the canonical Gaussian domain*, decoupling statistical significance from evidence magnitude.

We fuse the Z-scores using a weighted average determined by view confidence $\hat{w}_v = 1 - \hat{u}^v$. The joint Z-score $\boldsymbol{Z}^{joint}$ is computed as:

$$\boldsymbol{Z}^{joint} = \frac{\sum_{v=1}^{V} \hat{w}_v \boldsymbol{Z}^v}{\sqrt{\sum_{v=1}^{V} \hat{w}_v^2}}. \tag{9}$$

The denominator $\sqrt{\sum_{v=1}^{V} \hat{w}_v^2}$ is the standard variance stabilizer in Stouffer's method.

Finally, we interpret the joint Z-score $\boldsymbol{Z}^{joint}$ as the unified logit in the canonical evidence space. We recover the non-negative joint evidence via the activation function:

$$\hat{\boldsymbol{e}}^{joint} = a(\boldsymbol{Z}^{joint}). \tag{10}$$

This reconstructed evidence $\hat{\boldsymbol{e}}^{joint}$ forms the joint parameters $\hat{\boldsymbol{\alpha}}^{joint}$, completing the evidential framework. By fusing in the Z-space, SAEF ensures the combination is driven purely by the statistical significance of each view, rendering it robust to the magnitude fluctuations.

### 4.2. Optimization Objective

Given the training dataset $\mathcal{D} = \{(\boldsymbol{x}_i, \boldsymbol{y}_i)\}_{i=1}^{N}$, where $\boldsymbol{y}_i$ is the one-hot ground truth vector. For a specific view $v$, the neural network predicts the Dirichlet parameters $\boldsymbol{\alpha}_i^v = a(\boldsymbol{z}_i^v) + \mathbf{1}$. The training objective consists of two terms from the evidential learning paradigm (Han et al., 2022):

**Bayes Risk with Cross-Entropy.** We minimize the Bayes risk with respect to the cross-entropy loss over the induced Dirichlet distribution:

$$\begin{aligned}\mathcal{L}_{risk}(\boldsymbol{\alpha}_i^v, \boldsymbol{y}_i) &= \mathbb{E}_{\boldsymbol{p}_i \sim \text{Dir}(\boldsymbol{\alpha}_i^v)}\left[-\sum_{k=1}^{K} y_{ik} \log p_{ik}\right] \\ &= \sum_{k=1}^{K} y_{ik}\left(\psi(S_i^v) - \psi(\alpha_{ik}^v)\right),\end{aligned} \tag{11}$$

where $\psi(\cdot)$ denotes the digamma function, and $S_i^v = \sum_k \alpha_{ik}^v$ is the total Dirichlet strength.

**KL-Divergence Regularization.** To penalize false confidence on incorrect classes, we incorporate a Kullback-Leibler (KL) divergence term that forces the distribution of non-target classes towards a flat uniform distribution (Dirichlet prior $\boldsymbol{\alpha}_0 = \mathbf{1}$). Let $\tilde{\boldsymbol{\alpha}}_i^v = \boldsymbol{y}_i + (1 - \boldsymbol{y}_i) \odot \boldsymbol{\alpha}_i^v$ be

the rectified parameters removing the ground truth evidence. The regularization is defined as:

$$\mathcal{L}_{KL}(\boldsymbol{\alpha}_i^v) = KL\left[\text{Dir}(\tilde{\boldsymbol{\alpha}}_i^v) \parallel \text{Dir}(\mathbf{1})\right]. \tag{12}$$

**Total Objective.** The final objective is the summation of losses across all $V$ views:

$$\mathcal{L}_{total} = \sum_{v=1}^{V}\sum_{i=1}^{N}\left(\mathcal{L}_{risk}(\boldsymbol{\alpha}_i^v, \boldsymbol{y}_i) + \lambda_t \mathcal{L}_{KL}(\boldsymbol{\alpha}_i^v)\right), \tag{13}$$

where $\lambda_t = \min(1, t/T_{warm})$ is an annealing coefficient to linearly increase the regularization weight over the first $T_{warm}$ epochs.

## 5. Theoretical Analysis

In this section, we provide a rigorous analysis of SAEF, proving that it guarantees invariance to nuisance scale factors and robustness against asymmetric dominance, effectively resolving the vulnerabilities identified in Sec. 3.3. Furthermore, we analyze its statistical efficacy in suppressing noisy views. (Proofs in Appendix B.2)).

**Theorem 5.1** (**Global Scale Invariance**). *Let $\mathcal{F}_{SAEF}(\cdot)$ denote the fusion function of SAEF. Consider an input logit set $\mathcal{Z} = \{\boldsymbol{z}^v\}_{v=1}^{V}$ subject to an arbitrary scaling factor $1/T > 0$. Provided that the scaled signal strength exceeds the noise floor (i.e., $\frac{1}{T}\sigma(\boldsymbol{z}^v) > \epsilon$), SAEF is strictly invariant to global scaling:*

$$\mathcal{F}_{SAEF}\left(\left\{\frac{1}{T}\boldsymbol{z}^v\right\}_{v=1}^{V}\right) = \mathcal{F}_{SAEF}(\{\boldsymbol{z}^v\}_{v=1}^{V}). \tag{14}$$

This invariance formally resolves the *Evidential Scale Ambiguity* defined in Definition 3.1 since the view-specific uncertainty is derived from these standardized logits, ensuring uncertainty is decoupled from temperature $T$.

**Theorem 5.2** (**Robustness against Asymmetric Dominance**). *Consider a conflict fusion scenario with a reliable view $v_{rel}$ and an inflated conflicting view $v_{inf}$ scaled by $1/T$. Let $\boldsymbol{p}^{joint}$ be the fused probability distribution. For standard strategies (Sum, Mean, Weighted, DS), the fusion result is hijacked by the inflated view as $T \to 0$ (Amplification):*

$$\lim_{T \to 0} \boldsymbol{p}_{Baselines}^{joint} = \boldsymbol{p}^{v_{inf}}. \tag{15}$$

*In contrast, SAEF is strictly invariant to the asymmetric inflation, ensuring the predicted probability remains anchored to the true consensus of the unperturbed views:*

$$\boldsymbol{p}_{SAEF}^{joint}(\boldsymbol{z}^{v_{rel}}, \frac{1}{T}\boldsymbol{z}^{v_{inf}}) \equiv \boldsymbol{p}_{SAEF}^{joint}(\boldsymbol{z}^{v_{rel}}, \boldsymbol{z}^{v_{inf}}). \tag{16}$$

Consequently, SAEF prevents the *False Dominance* phenomenon identified in Corollary 3.3, ensuring the fusion

*Table 1.* **Comparative evaluation on standard benchmarks**. We report Accuracy (ACC) and AUROC across four datasets.

| Group | Method | AVE | | SUNRGBD | | CheXpert | | MURA | |
|---|---|---|---|---|---|---|---|---|---|
| | | ACC | AUROC | ACC | AUROC | ACC | AUROC | ACC | AUROC |
| *DS Fusion* | TMC (Han et al., 2022) | $82.42_{\pm1.54}$ | $98.22_{\pm0.34}$ | $45.08_{\pm6.99}$ | $83.71_{\pm1.23}$ | $44.13_{\pm3.08}$ | $73.22_{\pm1.11}$ | $78.21_{\pm0.24}$ | $83.87_{\pm0.42}$ |
| | TEF (Liang et al., 2025a) | $82.51_{\pm2.11}$ | $98.03_{\pm1.00}$ | $45.15_{\pm7.54}$ | $83.80_{\pm1.78}$ | $43.98_{\pm4.12}$ | $73.67_{\pm1.57}$ | $78.00_{\pm0.33}$ | $83.12_{\pm0.45}$ |
| *Sum Fusion* | TMDOA (Liu et al., 2022) | $82.40_{\pm1.24}$ | $98.18_{\pm0.12}$ | $45.93_{\pm7.01}$ | $83.72_{\pm1.22}$ | $44.45_{\pm3.44}$ | $73.39_{\pm1.04}$ | $78.37_{\pm0.35}$ | $83.94_{\pm0.36}$ |
| | RAR (Liu et al., 2025b) | $82.63_{\pm1.03}$ | $98.10_{\pm0.35}$ | $45.73_{\pm8.32}$ | $83.75_{\pm1.08}$ | $44.32_{\pm2.89}$ | $73.54_{\pm1.42}$ | $78.44_{\pm0.21}$ | $\mathbf{84.05}_{\pm0.30}$ |
| *Mean Fusion* | RCML (Xu et al., 2024) | $82.32_{\pm1.07}$ | $98.21_{\pm0.03}$ | $45.73_{\pm7.55}$ | $83.63_{\pm1.44}$ | $44.21_{\pm4.02}$ | $73.34_{\pm1.28}$ | $78.21_{\pm0.21}$ | $83.82_{\pm0.37}$ |
| | TMCEK (Liang et al., 2025b) | $82.34_{\pm1.13}$ | $98.27_{\pm0.10}$ | $45.69_{\pm7.26}$ | $83.42_{\pm1.09}$ | $44.40_{\pm3.87}$ | $73.38_{\pm1.23}$ | $78.22_{\pm0.18}$ | $83.90_{\pm0.56}$ |
| *Weighted Fusion* | SAEML (Xu et al., 2025) | $82.60_{\pm1.64}$ | $98.01_{\pm0.21}$ | $46.03_{\pm7.38}$ | $82.50_{\pm1.42}$ | $44.02_{\pm3.82}$ | $73.51_{\pm1.26}$ | $78.32_{\pm0.37}$ | $83.97_{\pm0.37}$ |
| **Ours** | **SAEF** | $\mathbf{82.84}_{\pm1.84}$ | $\mathbf{98.28}_{\pm0.32}$ | $\mathbf{48.17}_{\pm4.04}$ | $\mathbf{84.38}_{\pm2.20}$ | $\mathbf{44.97}_{\pm5.32}$ | $\mathbf{73.82}_{\pm1.15}$ | $\mathbf{78.97}_{\pm0.52}$ | $83.99_{\pm0.40}$ |

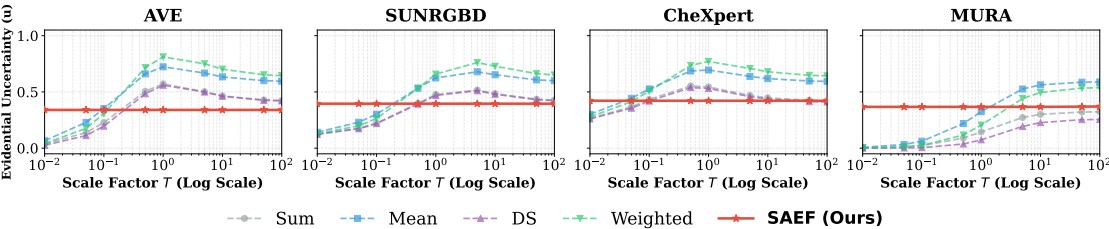

*Figure 2.* **Verification of Global Scale Invariance (Theorem 5.1).** We evaluate the stability of Evidential Uncertainty ($u$) under varying global scaling factors $T$. Specifically, for SAEF, we apply a fixed calibration scale of 5 when computing uncertainty, which does not affect invariance to $T$. While baselines (dashed lines) degenerate into False Confidence ($u \to 0$) or False Ignorance ($u \to u_{max}$) due to scale artifacts, SAEF (solid red line) maintains a stable uncertainty profile, validating that evidential confidence is decoupled from feature magnitude.

result remains determined by the semantic consensus rather than numerical magnitude ($T \to 0$).

**Theorem 5.3 (Noise Suppression via Stouffer Fusion).** *Consider a reliable view $v_{rel}$ and a noisy view $v_{noise}$ characterized by high evidence vacuity ($\hat{u}_{noise} \approx u_{max}$). In SAEF, the contribution of the noisy view to the joint decision is strongly down-weighted ($\hat{w}_{noise} = 1 - \hat{u}_{noise} \approx 0$) after standardization, irrespective of its original logit magnitude, ensuring the fusion result is dominated by the reliable view.*

This guarantees that vacuous (or statistically uninformative) views do not exhibit the *False Confidence* (Definition 3.1) observed in standard fusion, as their contribution is suppressed via the confidence weighting mechanism rather than amplified by raw logit scale.

## 6. Experiments

### 6.1. Experimental Setup

**Datasets.** We conduct experiments on four diverse multi-view datasets spanning medical diagnostics, scene understanding, and audio-visual classification: **SunRGBD** (Song et al., 2015), **AVE** (Tian et al., 2018), **CheXpert** (Irvin et al., 2019), and **MURA** (Rajpurkar et al., 2017). These benchmarks present varying challenges in terms of data scale ($4,143$ to $14,656$ samples), class cardinality (binary to 28 classes), and view configurations. Details are provided in Appendix C.1.

**Implementation Details.** We implement SAEF using

PyTorch on an NVIDIA RTX 4090 GPU. For vision-based datasets (SunRGBD, CheXpert, MURA), we employ ResNet-18 (He et al., 2016) pre-trained on ImageNet as the backbone. For AVE, we utilize officially provided pre-extracted features following (Tian et al., 2018). The models are trained using the Adam optimizer with a learning rate of $1e^{-3}$ and cosine annealing. Softplus is used for activation. Regarding SAEF-specific hyperparameters, we set the noise floor $\epsilon = 10^{-2}$ and the canonical scale $\beta = 5.0$ to ensure robust standardization. We simulate scale perturbations by varying the temperature $T \in \{0.01, 0.05, 0.1, 0.5, 1.0, 5.0, 10.0, 50.0, 100.0\}$. Details are provided in Appendix C.2. To quantify performance stability, we report the mean and standard deviation across 5 cross-validation folds for all experiments.

### 6.2. Comparative Evaluation

We compare SAEF against seven state-of-the-art methods across four canonical fusion paradigms. As shown in Table 1, SAEF consistently achieves the **best classification accuracy across all four benchmarks**, validating its superiority over magnitude-dependent baselines. Notably, on the challenging **SUNRGBD** dataset (RGB-D), SAEF surpasses the runner-up (SAEML) by a significant margin of **2.14%**. This verifies that our scale-invariant mechanism effectively unlocks the complementarity of heterogeneous modalities (RGB vs. Depth) where scale discrepancies are inherently severe. Furthermore, SAEF maintains state-of-the-art failure detection capabilities (AUROC),

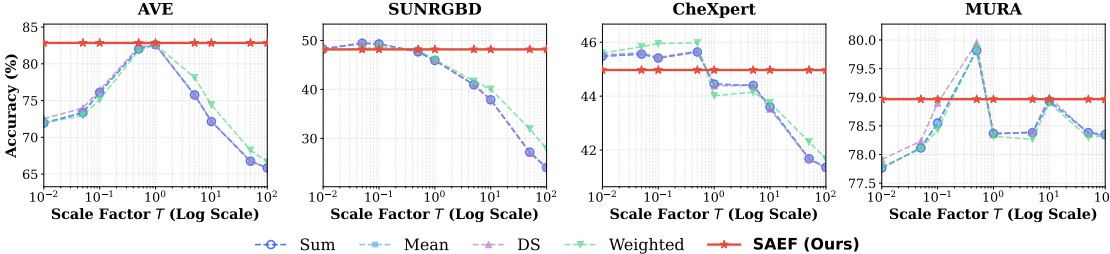

*Figure 3.* **Sensitivity Analysis under Asymmetric View Scaling (Theorem 5.2).** We simulate view imbalance by scaling the logits of a single view ($T \in [10^{-2}, 10^2]$). Baselines exhibit an "Inverted-V" collapse: amplification ($T < 1$) leads to single-view dominance, while suppression ($T > 1$) causes information loss. SAEF (solid red line) remains invariant, preserving multi-view complementarity.

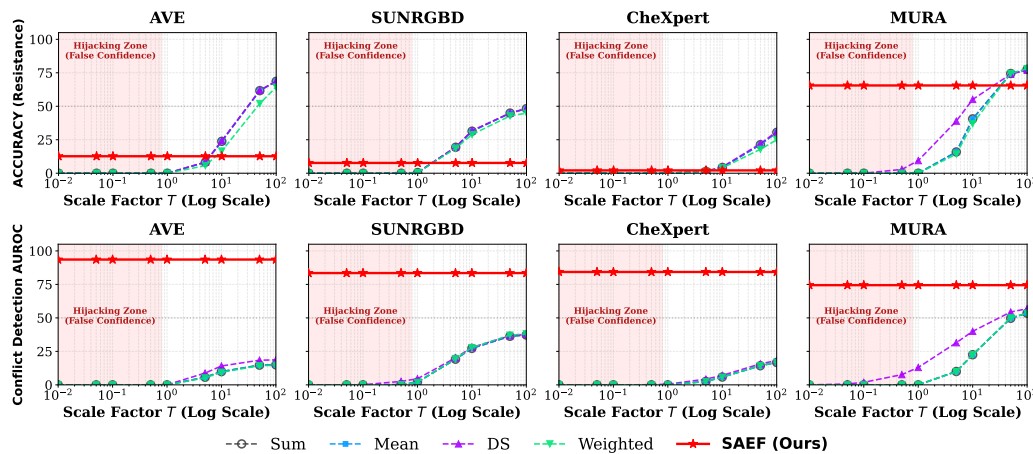

*Figure 4.* **Defense against Semantic Hijacking (Conflict Scenario).** We simulate a "confident lie" by injecting a conflicting view with high-magnitude logits ($T < 1$). **Top (Accuracy):** Baselines collapse to 0% accuracy, succumbing to *False Dominance*. SAEF resists total collapse, retaining partial information from the clean view. **Bottom (Detection AUROC):** Baselines exhibit *False Confidence* (AUROC $\approx 0$), failing to detect the attack. SAEF successfully flags the conflict with high uncertainty (High AUROC), acting as a reliable fail-safe.

ranking **1st** on AVE, SUNRGBD, and CheXpert. Even on the highly saturated MURA benchmark, it achieves the highest accuracy ($78.97\%$) with competitive reliability. These results confirm that SAEF enhances decision accuracy without compromising the semantic confidence required for safety-critical tasks.

### 6.3. Empirical Verification of Theoretical Guarantees

**1) Verification of Global Invariance:** We first validate **Theorem 5.1** by applying a global scaling factor $T$ to all input views and monitoring the evidential uncertainty ($u$) of the fused prediction. As shown in Figure 2, canonical fusion paradigms exhibit the *Evidential Scale Ambiguity* defined in Definition 3.1. Specifically, in the amplification regime ($T < 1$), the uncertainty of baselines collapses to zero ($u \to 0$), indicating *False Confidence* driven solely by inflated magnitudes. Conversely, suppression ($T > 1$) forces the model towards *False Ignorance* ($u \to u_{max}$). In contrast, SAEF maintains a nearly constant uncertainty profile across the tested spectrum. This empirically confirms

that SAEF successfully decouples semantic confidence from nuisance scale factors, ensuring the system's epistemic state remains stable against domain shifts.

**2) Robustness under Asymmetric Scaling:** To verify **Theorem 5.2**, we simulate an asymmetric scale imbalance where one view is scaled by $\boldsymbol{z}^v / T$ while the other remains fixed. We analyze the classification accuracy as a proxy for the stability of the fused probability distribution $\boldsymbol{p}^{joint}$. Figure 3 presents the results: Standard strategies exhibit a characteristic "Inverted-V" failure pattern. As predicted, amplification ($T \to 0$) causes the inflated view to mathematically hijack the fusion (**Validation of** Eq. (15)), leading to performance degradation towards single-view levels, while suppression ($T \to \infty$) results in information loss. In contrast, consistent with our theoretical derivation (**Validation of** Eq. (16)), SAEF demonstrates strict invariance to asymmetric scaling. The accuracy curve remains flat, confirming that the fused decision is anchored to the consensus of the underlying distributions rather than the loudest magnitude. This effectively prevents the *False Dominance* phenomenon

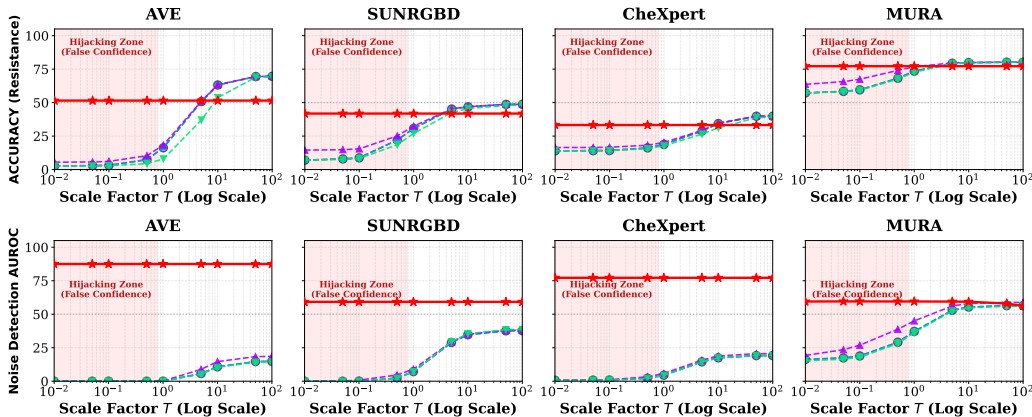

*Figure 5.* **Robustness against Sensor Noise (Empirical evaluation of Theorem 5.3).** We replace one view with scaled Gaussian noise. **Top (Accuracy):** Baselines treat amplified noise as valid signal, causing accuracy to plummet. SAEF remains stable, effectively filtering out the noise. **Bottom (Detection AUROC):** While baselines assign high confidence to noise (False Confidence), SAEF assigns high vacuity to corrupted inputs (High AUROC for $u$), validating its noise rejection capability.

*Table 2.* **Sensitivity Analysis of the Canonical Scale Factor $\beta$ (AVE Dataset).** The minimal variation (bottom row) confirms that SAEF maintains robust Accuracy, Conflict Detection, and Noise Rejection capabilities regardless of hyperparameter changes.

| Scale Factor $\beta$ | Clean Accuracy (ACC %) | Conflict Detection (AUROC %) | Noise Rejection (AUROC %) |
|---|---|---|---|
| 1 | $83.66 \pm 2.09$ | $97.45 \pm 0.83$ | $84.59 \pm 1.49$ |
| 2 | $83.28 \pm 1.82$ | $94.60 \pm 1.18$ | $85.20 \pm 1.96$ |
| 3 | $82.94 \pm 1.97$ | $93.06 \pm 1.37$ | $85.66 \pm 2.16$ |
| 5 | $82.84 \pm 1.84$ | $93.55 \pm 1.40$ | $86.25 \pm 2.30$ |
| 7 | $82.65 \pm 1.72$ | $94.82 \pm 1.29$ | $86.53 \pm 2.31$ |
| 10 | $82.51 \pm 1.71$ | $95.79 \pm 1.18$ | $86.67 \pm 2.28$ |
| 15 | $82.51 \pm 1.73$ | $95.95 \pm 1.17$ | $86.64 \pm 2.24$ |
| 20 | $82.41 \pm 1.73$ | $95.54 \pm 1.23$ | $86.57 \pm 2.21$ |
| **Variation** | $82.85 \pm \mathbf{0.41}$ | $95.09 \pm \mathbf{1.31}$ | $86.01 \pm \mathbf{0.73}$ |

*Table 3.* **Sensitivity Analysis of the Noise Floor $\epsilon$.** We evaluate the impact of $\epsilon$ across orders of magnitude with $T = 0.1$. The results remain stable for $\epsilon \leq 0.1$, confirming that the gating mechanism targets only effectively silenced views without interfering with informative features.

| Noise Floor $\epsilon$ | Clean Accuracy (ACC %) | Conflict Detection (AUROC %) | Noise Rejection (AUROC %) |
|---|---|---|---|
| 1e-5 | $82.84 \pm 1.84$ | $93.55 \pm 1.40$ | $87.40 \pm 2.00$ |
| 1e-4 | $82.84 \pm 1.84$ | $93.55 \pm 1.40$ | $87.40 \pm 2.00$ |
| 1e-3 | $82.84 \pm 1.84$ | $93.55 \pm 1.40$ | $87.40 \pm 2.00$ |
| 0.01 | $82.84 \pm 1.84$ | $93.55 \pm 1.40$ | $87.40 \pm 2.00$ |
| 0.1 | $82.84 \pm 1.84$ | $93.55 \pm 1.40$ | $87.40 \pm 2.00$ |
| 0.5 | $82.84 \pm 1.84$ | $93.60 \pm 1.33$ | $87.41 \pm 1.98$ |
| 1.0 | $83.08 \pm 1.46$ | $94.34 \pm 0.85$ | $87.16 \pm 2.65$ |
| **Variation** | $82.87 \pm \mathbf{0.08}$ | $93.67 \pm \mathbf{0.27}$ | $87.37 \pm \mathbf{0.08}$ |

(Corollary 3.3) and preserves multi-view complementarity.

### 6.4. Robustness and Safety Analysis

**1) Conflict Resistance and Detection (Semantic Hijacking):** To evaluate safety under adversarial conditions, we synthesize a *Semantic Hijacking* scenario. One view is corrupted by assigning high-magnitude logits to an incorrect shifted label $(y + 1) \mod K$, with the shifted class fixed at 100 and then scaled by $1/T$. We measure Accuracy to assess resistance and AUROC (using uncertainty $u$) to assess detection capability. As presented in Figure 4, standard fusion strategies suffer catastrophic failure in the Hijacking Zone ($T < 1$): Consistent with Corollary 3.3, baselines are mathematically forced to predict the wrong class due to magnitude dominance (Accuracy $\approx 0\%$). Crucially, they exhibit extreme *False Confidence* (AUROC $\approx 0\%$), failing to detect the conflict entirely. In contrast, SAEF demonstrates robust safety properties. This confirms that even when

semantic ambiguity exists, SAEF correctly flags the sample as highly uncertain rather than making a silent failure.

**2) Noise Robustness and Detection (Sensor Failure):** We further empirically evaluate the noise-suppression behavior by simulating sensor failure, where one view is replaced by Gaussian noise sampled from $\mathcal{N}(0, 10^2 I)$ and subjected to scaling by $1/T$. This tests the system's ability to distinguish "loud noise" from "valid signal". Figure 5 illustrates the results: Standard paradigms lack intrinsic mechanisms to distinguish signal strength from variance. Consequently, high-magnitude noise numerically overwhelms the clean view, causing accuracy to plummet to near-zero levels. The near-zero AUROC indicates that baselines confidently accept sensor glitches as valid predictions. In contrast, SAEF empirically supports the *Noise Suppression* guarantee: **(i) Signal Preservation:** The accuracy curve remains flat, confirming that SAEF bounds the effect of noise variance, preventing it from dominating the fusion. **(ii)**

*Table 4.* **Ablation Study on Normalization and Fusion Strategies (AVE Dataset).** We evaluate the impact of each component on Accuracy (Clean Data), Conflict Detection (Semantic Hijacking), and Noise Rejection (Sensor Failure) with $T = 1$. **Key Findings:** 1) Without Standardization (No Norm), the model completely fails to detect conflicts (AUROC $\approx 14\%$). 2) Without Stouffer Fusion (e.g., Sum/Mean), the model fails to reject noise (AUROC $\approx 27\%$), even if standardization is applied. SAEF is the only configuration that ensures safety across all scenarios.

| Method / Configuration | Norm Strategy | Fusion Strategy | Clean ACC (%) | Conflict AUROC (%) | Noise AUROC (%) |
|---|---|---|---|---|---|
| No Norm (Baseline) | None | Stouffer | $83.37 \pm 1.41$ | $14.06 \pm 6.61$ | $14.54 \pm 6.75$ |
| Batch Norm | Batch | Stouffer | $83.57 \pm 1.24$ | $88.19 \pm 1.29$ | $86.02 \pm 2.57$ |
| Layer Norm | Layer | Stouffer | $82.84 \pm 1.84$ | $\mathbf{93.55} \pm 1.40$ | $86.92 \pm 2.51$ |
| **SAEF (Ours)** | **Instance** | **Stouffer** | $82.84 \pm 1.84$ | $\mathbf{93.55} \pm 1.40$ | $\mathbf{87.03} \pm 2.22$ |
| SAEF + Sum | Instance | Sum | $83.57 \pm 1.81$ | $90.66 \pm 0.48$ | $27.21 \pm 3.32$ |
| SAEF + Mean | Instance | Mean | $\mathbf{83.66} \pm 1.71$ | $93.51 \pm 0.42$ | $28.19 \pm 3.54$ |
| SAEF + Max | Instance | Max | $80.10 \pm 1.87$ | $63.58 \pm 3.98$ | $24.46 \pm 3.67$ |

*Table 5.* **Runtime Analysis.** Values represent the average inference time per sample (in milliseconds) across 5 folds.

| Method | AVE | CheXpert | SUNRGBD | MURA |
|---|---|---|---|---|
| DS | 0.42 | 2.60 | 4.83 | 5.43 |
| Sum | 0.49 | 2.53 | 4.98 | 5.44 |
| Mean | 0.39 | 2.58 | 4.95 | 5.32 |
| Weighted | 0.41 | 2.56 | 4.83 | 5.48 |
| **Ours** | 0.45 | 2.54 | 4.94 | 5.45 |

**Noise Rejection:** The contribution of the noisy view is strongly suppressed, resulting in high AUROC scores. This demonstrates SAEF's capability to detect sensor anomalies under the considered perturbation protocol.

### 6.5. Parameter Sensitivity

**1) Sensitivity Analysis of canonical scale factor $\beta$:** We evaluate SAEF on the AVE dataset across a wide range of $\beta \in [1, 20]$. We report three key metrics: **(i) Clean Accuracy (ACC):** To measure performance stability on standard data. **(ii) Conflict Detection (AUROC):** To assess safety under semantic hijacking attacks ($T = 0.1$). **(iii) Noise Rejection (AUROC):** To evaluate the ability to filter out sensor noise ($T = 0.1$). The results are summarized in Table 2. SAEF exhibits exceptional stability across the entire operating range. The performance variation across different $\beta$ settings is negligible, with standard deviations of **0.41%** (Accuracy), **1.31%** (Conflict AUROC), and **0.73%** (Noise AUROC). Notably, even at extreme values (e.g., $\beta = 1$ or $\beta = 20$), the model maintains high safety standards across both conflict and noise scenarios.

**2) Sensitivity Analysis of Noise Floor $\epsilon$:** We analyze the sensitivity of the noise floor $\epsilon$ (Eq. (7)), which acts as a lower bound for variance gating. As shown in Table 3, the performance remains strictly invariant for $\epsilon \leq 0.1$. This stability is theoretically expected: for informative views, the feature standard deviation $\sigma$ typically exceeds the noise floor ($\sigma \gg \epsilon$), rendering the gating mechanism inactive.

The mechanism only activates to clamp variance when the view is effectively silenced or dominated by numerical noise, thereby ensuring safety without affecting standard inference.

### 6.6. Ablation Study

As shown in Table 4, the results reveal two critical findings: **(i) Necessity of Standardization:** Removing instance-wise standardization causes the conflict detection AUROC to collapse from $93.55\%$ to $14.06\%$, exposing the model to extreme false confidence. **(ii) Necessity of Stouffer Fusion:** Replacing Stouffer fusion with standard Mean/Sum operations results in a noise rejection AUROC of only $\approx 28\%$, failing to filter out sensor anomalies. These findings confirm that both components are indispensable for safety. Detailed component analysis can be found in **Appendix C.3**.

### 6.7. Runtime Analysis

We conducted a runtime test on a single NVIDIA RTX 4090 GPU. Table 5 reports the average inference time per sample (in milliseconds) across the 5 folds. As empirically shown, the computational overhead introduced by SAEF is practically negligible. For instance, the latency difference compared to the standard Mean fusion is bounded within $\sim 0.1$ms, demonstrating the highly lightweight nature of our proposed framework.

## 7. Conclusion

We identify the Scale Mismatch Problem in evidential fusion, where sensitivity to logit scales exposes systems to false dominance. We propose Scale-Invariant Evidential Fusion (SAEF), employing instance-wise standardization to decouple numerical magnitude from semantic reliability. Theoretically, SAEF ensures strict scale invariance and noise suppression. Comprehensive evaluations demonstrate that SAEF establishes a new state-of-the-art in classification accuracy while functioning as a robust fail-safe against semantic conflicts and sensor noise.

## Acknowledgments

This work was supported by the National Natural Science Foundation of China (Nos. 62472315, 62476165).

## Impact Statement

This paper presents work whose goal is to advance the field of Machine Learning. There are many potential societal consequences of our work, none which we feel must be specifically highlighted here.

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

# A. Derivations of Evidential Fusion Strategies

In this section, we provide the detailed formulations and derivations for the fusion strategies discussed in the main text. We begin by establishing the fundamental relationship between evidence and belief mass in Evidential Deep Learning (EDL).

## A.1. Preliminaries: The Evidence-Belief Identity

For a view $v$ with $K$ classes, let $e_k^v$, $b_k^v$, and $u^v$ denote the evidence, belief mass, and uncertainty, respectively. The Dirichlet strength is defined as $S^v = \sum_{k=1}^{K} e_k^v + K = K/u^v$. The fundamental relationship between evidence and belief is given by:

$$e_k^v = S^v b_k^v = \frac{K}{u^v} b_k^v. \tag{17}$$

This yields the critical identity that links evidential magnitude to belief-uncertainty ratio:

$$\frac{b_k^v}{u^v} = \frac{e_k^v}{K}. \tag{18}$$

## A.2. Linear Fusion Strategies

We categorize Sum, Mean, and Weighted fusion as *linear strategies*, as they operate through linear combinations of the evidence vectors.

**1. Sum Fusion.** As formulated in (Liu et al., 2022), Sum fusion assumes that evidence from distinct views is additive. It directly accumulates the evidence vectors:

$$\boldsymbol{e}_{Sum}^{joint} = \sum_{v=1}^{V} \boldsymbol{e}^v. \tag{19}$$

**2. Mean Fusion (Derivation from Subjective Logic).** The Mean fusion corresponds to the **Average Operator** in Subjective Logic (Xu et al., 2024). While originally defined in the belief space, we prove here that it simplifies to an arithmetic mean in the evidence space.

*Definition:* For two views $A$ and $B$, the fused belief $b_k^{joint}$ and uncertainty $u^{joint}$ are defined as:

$$b_k^{joint} = \frac{b_k^A u^B + b_k^B u^A}{u^A + u^B}, \tag{20}$$

$$u^{joint} = \frac{2u^A u^B}{u^A + u^B}. \tag{21}$$

*Derivation:* To obtain the fused evidence $e_k^{joint}$, we substitute Eq. (20) and Eq. (21) into the standard definition $e_k = \frac{K}{u} b_k$:

$$\begin{aligned}
e_k^{joint} &= \frac{K}{u^{joint}} \cdot b_k^{joint} \\
&= K \cdot \underbrace{\left( \frac{u^A + u^B}{2u^A u^B} \right)}_{1/u^{joint}} \cdot \underbrace{\left( \frac{b_k^A u^B + b_k^B u^A}{u^A + u^B} \right)}_{b_k^{joint}} \\
&= \frac{K}{2u^A u^B} \left( b_k^A u^B + b_k^B u^A \right) \quad \text{(The terms } (u^A + u^B) \text{ cancel out)} \\
&= \frac{K}{2} \left( \frac{b_k^A u^B}{u^A u^B} + \frac{b_k^B u^A}{u^A u^B} \right) \\
&= \frac{K}{2} \left( \frac{b_k^A}{u^A} + \frac{b_k^B}{u^B} \right).
\end{aligned}$$

Applying the identity from Eq. (18) ($\frac{b_k}{u} = \frac{e_k}{K}$), we obtain:

$$
\begin{aligned}
e_k^{joint} &= \frac{K}{2} \left( \frac{e_k^A}{K} + \frac{e_k^B}{K} \right) \\
&= \frac{1}{2} \left( e_k^A + e_k^B \right).
\end{aligned}
\tag{22}
$$

This derivation proves that the complex belief aggregation rules in Subjective Logic are mathematically equivalent to linear averaging at the evidence level.

**3. Weighted Fusion.** As derived in (Xu et al., 2025), Weighted fusion introduces importance weights $\tilde{w}_v$ based on view confidence:

$$
\boldsymbol{e}_{Weighted}^{joint} = \sum_{v=1}^{V} \tilde{w}_v \boldsymbol{e}^v, \quad \text{where } \tilde{w}_v = \frac{1 - u^v}{\sum_{j=1}^{V}(1 - u^j)}.
\tag{23}
$$

Since the weights $\tilde{w}_v$ are scalar coefficients, this strategy remains a linear combination of the input evidence.

### A.3. Non-linear Strategy: Dempster-Shafer Fusion

In contrast to the linear strategies above, Dempster-Shafer (DS) fusion involves multiplicative interactions between views. Referring to the unnormalized expansion derived in (Lu et al., 2025), the joint evidence for two views is given by:

$$
\boldsymbol{e}_{DS}^{joint} = \boldsymbol{e}^1 + \boldsymbol{e}^2 + \frac{1}{K}(\boldsymbol{e}^1 \odot \boldsymbol{e}^2).
\tag{24}
$$

The presence of the Hadamard product term $\boldsymbol{e}^1 \odot \boldsymbol{e}^2$ introduces a quadratic non-linearity, which is the structural cause of the higher-order scale sensitivity discussed in the main text.

## B. Proofs

### B.1. The Scale Mismatch Problem

#### Definition 3.1 (Evidential Scale Ambiguity)
*Definition* (**Evidential Scale Ambiguity**). A reliable uncertainty estimator should be invariant to affine transformations that preserve the semantic direction of predictions. However, for standard EDL models, the evidential uncertainty $u^v$ is strictly coupled with the temperature $T$:

$$
\begin{aligned}
\lim_{T \to 0} u^v(T) = 0 \quad &\text{(False Confidence)}, \\
\lim_{T \to \infty} u^v(T) = u_{\max} \quad &\text{(False Ignorance)}.
\end{aligned}
\tag{25}
$$

where $u_{\max}$ represents the activation-dependent maximum uncertainty capacity (e.g., $u_{\max} = 1$ for ReLU).

*Proof.* Recall that $u^v = K/S^v$ with Dirichlet strength $S^v = \sum e_k^v + K$. Since $a(\cdot)$ is monotonically increasing, $T \to 0$ drives $S^v \to \infty$, artificially vanishing uncertainty ($u^v \to 0$). Conversely, $T \to \infty$ drives inputs to zero. The evidence collapses to the activation bias (e.g., $a(0)$), forcing uncertainty to its achievable maximum $u_{\max}$ (i.e., the prior state), independent of semantic content. $\qquad\square$

#### Proposition 3.2 (Scale Sensitivity of Fusion Strategies)
*Proposition* (**Scale Sensitivity of Fusion Strategies**). Let $T$ be the temperature scaling factor applied to input logits. Assuming the activation function exhibits asymptotic homogeneity (i.e., $a(z/T) \approx \frac{1}{T}a(z)$ as $T \to 0$, strictly satisfied by ReLU), standard fusion strategies exhibit order-dependent sensitivity to $T$:

**1) Linear Sensitivity (Sum, Mean, Weighted):** These strategies propagate scale linearly. The magnitude of the fused evidence scales as $\mathcal{O}(1/T)$. Consequently, $T \to 0$ leads to linear inflation (False Confidence).

**2) Quadratic Sensitivity (DS):** Due to the multiplicative interaction term, DS fusion suffers from *quadratic sensitivity*, where the fused magnitude scales as $\mathcal{O}(1/T^2)$, causing explosive instability under global amplification.

*Proof.* We analyze the asymptotic behavior in the amplification regime ($T \to 0$), where the approximation $\tilde{e}^v(T) \approx \frac{1}{T} e^v$ holds (e.g., $\ln(1 + e^{z/T}) \approx z/T$). For *Linear strategies* (Sum/Mean), the fusion operation preserves linearity (e.g., $\|\sum \frac{1}{T} e^v\| = \frac{1}{T}\|\sum e^v\|$), implying $\mathcal{O}(1/T)$ sensitivity. For *Weighted Fusion*, since weights are normalized ($\sum \tilde{w}_v = 1$), the fused magnitude remains bounded by the convex combination of the scaled inputs, preserving $\mathcal{O}(1/T)$ linearity.

In contrast, for *DS Fusion*, considering the unnormalized evidence expansion derived in (Lu et al., 2025), the joint evidence behaves as:

$$\tilde{e}^{joint} \propto \frac{1}{T} \underbrace{(e^1 + e^2)}_{\text{Linear Term}} + \frac{1}{T^2} \underbrace{\frac{1}{K}(e^1 \odot e^2)}_{\text{Quadratic Interaction}} , \tag{26}$$

The presence of the product term $e^1 \odot e^2$ introduces a quadratic dependency on the scaling factor $1/T$. Under amplification ($T \to 0$), the quadratic interaction term ($\frac{1}{T^2}$) dominates the linear term, causing the fused evidence magnitude to explode significantly faster than linear approaches. This validates that DS fusion is structurally hypersensitive to scale perturbations, leading to rapid convergence to False Confidence. Conversely, under suppression ($T \to \infty$), the evidence collapses (to zero for ReLU or a constant bias for Softplus), rendering the interaction term structurally uninformative. $\square$

## B.2. Theoretical Analysis

### Theorem 5.1 (Global Scale Invariance)

*Theorem* (**Global Scale Invariance**). Let $\mathcal{F}_{SAEF}(\cdot)$ denote the fusion function of SAEF. Consider an input logit set $\mathcal{Z} = \{z^v\}_{v=1}^V$ subject to an arbitrary scaling factor $1/T > 0$. Provided that the scaled signal strength exceeds the noise floor (i.e., $\frac{1}{T}\sigma(z^v) > \epsilon$), SAEF is strictly invariant to global scaling:

$$\mathcal{F}_{SAEF}\left(\left\{\frac{1}{T}z^v\right\}_{v=1}^V\right) = \mathcal{F}_{SAEF}(\{z^v\}_{v=1}^V). \tag{27}$$

*Proof.* Under the condition $\sigma(\tilde{z}^v) = \sigma(z^v)/T > \epsilon$, the denominator in Eq. (7) is $\sigma(\tilde{z}^v)$. Let $\tilde{z}^v = \frac{1}{T}z^v$. Due to the linearity of expectation, the mean and standard deviation scale linearly: $\mu(\tilde{z}^v) = \frac{1}{T}\mu(z^v)$ and $\sigma(\tilde{z}^v) = \frac{1}{T}\sigma(z^v)$. Substituting these into the standardization formula:

$$\hat{\tilde{z}}^v = \beta \frac{\frac{1}{T}z^v - \frac{1}{T}\mu(z^v)}{\frac{1}{T}\sigma(z^v)} = \beta \frac{\frac{1}{T}(z^v - \mu(z^v))}{\frac{1}{T}\sigma(z^v)} = \hat{z}^v. \tag{28}$$

The scaling factor $1/T$ cancels out completely. **Critically, since the view-specific uncertainty is derived from these standardized logits** (i.e., $\hat{u}^v = K/(\sum a(\hat{z}^v) + K)$), **the invariance of $\hat{z}^v$ directly guarantees that $\hat{u}^v(T) \equiv \hat{u}^v(1)$.** Consequently, both the single-view uncertainty and the final fused result remain identical regardless of $T$. This formally resolves the *Evidential Scale Ambiguity* (Definition 3.1), proving that the uncertainty is now strictly decoupled from the nuisance temperature. $\square$

### Theorem 5.2 (Robustness against Asymmetric Dominance)

*Theorem* (**Robustness against Asymmetric Dominance**). Consider a conflict fusion scenario with a reliable view $v_{rel}$ and an inflated conflicting view $v_{inf}$ scaled by $1/T$. Let $p^{joint}$ be the fused probability distribution. For standard strategies (Sum, Mean, Weighted, DS), the fusion result is hijacked by the inflated view as $T \to 0$ (Amplification):

$$\lim_{T \to 0} p^{joint}_{Baselines} = p^{v_{inf}}. \tag{29}$$

In contrast, SAEF is strictly invariant to the asymmetric inflation, ensuring the predicted probability remains anchored to the true consensus of the unperturbed views:

$$p^{joint}_{SAEF}(z^{v_{rel}}, \frac{1}{T}z^{v_{inf}}) \equiv p^{joint}_{SAEF}(z^{v_{rel}}, z^{v_{inf}}). \tag{30}$$

*Proof.* We analyze the asymptotic behavior. For Sum, Mean, and DS fusion, as derived in Proposition 3.2, the output magnitude scales proportionally to the input ($\mathcal{O}(1/T)$ or $\mathcal{O}(1/T^2)$). As $T \to 0$, the evidence from $v_{inf}$ numerically drowns out $v_{rel}$. Even for confidence-weighted fusion, the weights depend on uncertainty $u \propto 1/S$. As $T \to 0$, the inflated

view exhibits ***False Confidence*** (Definition 3.1), causing $u_{inf} \to 0$ and assigning the inflated view an unjustifiably large confidence weight. Together with its inflated evidence magnitude, this suppresses the reliable view (where $\tilde{w}_{rel} \to 0$ relative to the inflated term), causing the fusion to collapse to $\boldsymbol{p}^{v_{inf}}$. In contrast, SAEF applies instance standardization *before* weight calculation. The factor $1/T$ cancels out in Eq. (28), meaning $\hat{\boldsymbol{z}}^{v_{inf}}(T) \equiv \hat{\boldsymbol{z}}^{v_{inf}}(1)$. Consequently, both the reconstructed evidence magnitude and the fusion weights remain invariant. The reliable view $v_{rel}$ retains its proper contribution, guaranteeing that the final probability distribution remains stable, effectively preventing the "hostile takeover" by the inflated view. $\square$

**Theorem 5.3 (Noise Suppression via Stouffer Fusion)**

*Theorem* (**Noise Suppression via Stouffer Fusion**). Consider a reliable view $v_{rel}$ and a noisy view $v_{noise}$ characterized by high evidence vacuity ($\hat{u}_{noise} \approx u_{max}$). In SAEF, the contribution of the noisy view to the joint decision is strongly down-weighted ($\hat{w}_{noise} = 1 - \hat{u}_{noise} \approx 0$) after standardization, irrespective of its original logit magnitude, ensuring the fusion result is dominated by the reliable view.

*Proof.* Consider a noisy view characterized by a lack of informative features (i.e., a nearly flat distribution where the intrinsic standard deviation $\sigma^v < \epsilon$).

First, we analyze the standardization step. The instance-wise mean $\mu^v$ removes any global shift, and due to the **Variance Gating** mechanism in Eq. (7), the denominator is clamped to $\epsilon$. Consequently, for a nearly flat logit vector, the standardized logits are bounded and close to zero:

$$\hat{\boldsymbol{z}}^v = \beta \frac{\boldsymbol{z}^v - \mu^v}{\epsilon} \approx \boldsymbol{0}. \tag{31}$$

This yields nearly uniform calibrated evidence, resulting in high evidential vacuity ($\hat{u}_{noise} \approx u_{max}$) and a small fusion weight ($\hat{w}_{noise} = 1 - \hat{u}_{noise} \approx 0$). Importantly, this behavior is determined by the standardized variation of the view rather than its original logit magnitude.

Next, we examine the Uncertainty-Weighted Stouffer Fusion. Recall the fusion formula:

$$\boldsymbol{Z}^{joint} = \frac{\hat{w}_{rel} \boldsymbol{Z}^{rel} + \hat{w}_{noise} \boldsymbol{Z}^{noise}}{\sqrt{\hat{w}_{rel}^2 + \hat{w}_{noise}^2}}. \tag{32}$$

When $\hat{w}_{noise}$ is small, the noisy contribution is suppressed, and the joint score is dominated by the reliable view. In the limiting case $\hat{w}_{noise} \to 0$, we obtain:

$$\lim_{\hat{w}_{noise} \to 0} \boldsymbol{Z}^{joint} = \frac{\hat{w}_{rel} \boldsymbol{Z}^{rel}}{|\hat{w}_{rel}|} = \boldsymbol{Z}^{rel}, \tag{33}$$

where $\hat{w}_{rel} > 0$ for an informative reliable view. This confirms that vacuous or statistically uninformative evidence is filtered out by the confidence weighting mechanism, preventing the noisy source from dominating the decision through raw magnitude. $\square$

### B.3. Computational Complexity

Despite introducing statistical rectification, SAEF maintains the same asymptotic complexity as elementary fusion strategies. Let $V$ be the number of views and $K$ be the number of classes. Standard strategies (Sum/Mean/Weighted) operate with $\mathcal{O}(VK)$. Although Dempster-Shafer fusion involves interaction terms, its closed-form recursive implementation also scales as $\mathcal{O}(VK)$. Similarly, SAEF operates strictly element-wise using scalar statistical moments and efficient mappings ($\Phi, \Phi^{-1}$), preserving the linear complexity $\mathcal{O}(VK)$.

## C. Experiment

### C.1. Dataset Details and Preprocessing

We provide detailed descriptions of the four datasets used in our evaluation, including the data curation and view construction strategies.

*Table 6.* **Summary of Datasets.** We evaluate SAEF across four diverse multi-view benchmarks. "CV" denotes the number of folds used for cross-validation on the development set.

| Dataset | Domain | Views ($V$) | Classes ($K$) | Total Samples | Test Set | Protocol |
|---|---|---|---|---|---|---|
| **SunRGBD** (Song et al., 2015) | Scene | 2 | 15 | 9,882 | 989 | 5-fold CV |
| **AVE** (Tian et al., 2018) | Audio-Visual | 2 | 28 | 4,143 | 415 | 5-fold CV |
| **CheXpert** (Irvin et al., 2019) | Medical | 2 | 7 | 5,621 | 1,529 | 5-fold CV |
| **MURA** (Rajpurkar et al., 2017) | Medical | 3 | 2 | 14,656 | 1,199 | 5-fold CV |

- **SunRGBD** (Song et al., 2015):A benchmark for indoor scene understanding containing synchronized RGB and Depth images ($V = 2$). The original dataset comprises 10,335 samples with significant class imbalance. We performed data cleaning by merging semantically similar scene descriptions into unified categories and removing tail classes with fewer than 50 samples. The final curated dataset consists of **9,882** samples distributed across **15** distinct scene classes.

- **Audio-Visual Event (AVE)** (Tian et al., 2018): A dataset for multi-modal event classification using video frames and audio tracks ($V = 2$). We utilize the supervised subset containing **4,143** samples covering **28** event categories (e.g., dog barking, violin playing). The visual and acoustic features serve as two heterogeneous views for fusion.

- **CheXpert** (Irvin et al., 2019): A large-scale chest radiograph dataset. To construct a rigorous multi-view benchmark, we filtered the dataset to select studies that strictly contain both Frontal and Lateral X-rays ($V = 2$). Furthermore, to eliminate label ambiguity during evaluation, we selected samples where both views share the same single-label diagnosis. This process resulted in **5,621** samples classified into **7** distinct pathologies.

- **MURA** (Rajpurkar et al., 2017): A musculoskeletal radiography dataset for binary abnormality detection. A significant challenge in MURA is the variable number of views per patient study. To unify the input configuration for batch training, we standardized each study to a fixed 3-view format ($V = 3$). For studies with fewer than 3 views, we applied random duplication; for those with more, we randomly truncated the excess views. The final dataset comprises **14,656** patient studies.

**Evaluation Protocol.** For all datasets, we adhere to a rigorous evaluation protocol to ensure statistical reliability. We employ a **5-fold cross-validation** scheme on the development set. Specifically, the data is randomly partitioned into 5 folds: in each run, 4 folds constitute the training set, while 1 fold serves as the validation set for hyperparameter tuning and model selection. Crucially, to prevent information leakage and ensure unbiased evaluation, all final results (Accuracy, AUROC, Uncertainty score) are reported on a **separate, held-out test set** that is never seen during the training or validation phases. We report the mean and standard deviation across the 5 folds to quantify performance stability.

## C.2. Detailed Implementation

In this section, we provide the complete details of our experimental setting to facilitate reproducibility, covering network architectures, training dynamics, and hyperparameter configurations.

**Network Architectures.** For the image-based datasets (**SUNRGBD**, **MURA**, and **CheXpert**), we employ **ResNet-18** (He et al., 2016) pre-trained on ImageNet as the unified backbone. We extract high-level feature representations from the final pooling layer of each view (dimension $d = 512$). To ensure fair comparison and strictly follow the protocol of prior arts (Tian et al., 2018), for the **AVE** dataset, we utilize the officially provided pre-extracted visual and audio features without additional fine-tuning. For all datasets, the extracted features are mapped to the Dirichlet parameters via a view-specific non-linear projection head. This head consists of two fully connected layers with ReLU activation, mapping the feature dimension to the number of classes $K$, followed by the SAEF module.

**Training Strategies.** All models are implemented in PyTorch and trained on a single **NVIDIA GeForce RTX 4090 GPU**. We adopt a decoupled end-to-end training strategy (as detailed in Sec. 4.2) using the **Adam optimizer**.

- **Learning Rate:** We use a standard initial learning rate of $1 \times 10^{-3}$, adjusted via a cosine annealing scheduler that decays the rate to $1 \times 10^{-6}$ over the course of training.

- **Regularization:** We apply a weight decay of $1 \times 10^{-4}$ to prevent overfitting and use gradient clipping with a max norm of 1.0 to ensure stability during evidential updates.

- **Batch & Epochs:** We set the batch size to 128 and train the models for $T_{num} = 100$ epochs. The annealing coefficient $\lambda_t$ for the KL-divergence term is linearly increased from 0 to 1 over the epochs, i.e., $\lambda_t = \min(1, t/T_{num})$.

**Hyperparameters for SAEF.** Unless otherwise specified in the sensitive studies, we use the following default hyperparameters for Scale-Invariant Evidential Fusion:

- **Noise Floor ($\epsilon$):** Set to $10^{-2}$. This acts as a variance gate to suppress uninformative variations (pure noise) from being amplified during standardization.

- **Canonical Scale ($\beta$):** Set to 5.0. This ensures that the standardized logits fall into the informative, linear regime of the Softplus activation function, preventing the uncertainty plateau issue.

- **Scale Perturbation ($T$):** To evaluate robustness, we simulate diverse scale mismatches by dividing the view-specific logits $\mathbf{z}$ by a factor $T$. The test set covers a wide spectrum $T \in [0.01, 0.05, 0.1, 0.5, 1.0, 5.0, 10.0, 50.0, 100.0]$, representing conditions from severe signal amplification ($T \ll 1$) to extreme suppression ($T \gg 1$).

### C.3. Detailed Ablation Study

In the main paper, we briefly discussed the necessity of SAEF's components. Here, we present the comprehensive ablation study on the AVE dataset (Table 4), dissecting the specific contributions of Normalization Strategies and Fusion Mechanisms.

### 1. Why Instance-wise Standardization?

We compared SAEF against No Norm, Batch Norm (BN), and Layer Norm (LN).

- **No Norm:** As shown in the first row, without standardization, the Conflict AUROC is abysmal ($14.06\%$). This proves that raw logit magnitude acts as a confounding variable, misleading the model into high confidence during semantic hijacking.

- **BN vs. SAEF:** While BN improves robustness, it relies on batch statistics. SAEF achieves higher stability ($93.55\%$) by strictly enforcing instance-level constraints.

### 2. Why Stouffer Fusion?

Even with standardization enabled, the choice of fusion mechanism is critical for noise robustness.

- **Linear Fusion Failure:** Configurations like "SAEF + Sum" or "SAEF + Mean" achieve high Conflict AUROC but fail spectacularly on Noise Rejection ($\sim 27 - 28\%$). This indicates that linear aggregation allows high-variance noise (even if standardized) to pollute the joint prediction.

- **Stouffer's Filter:** SAEF (using Stouffer Fusion) is the only configuration that maintains high Noise AUROC ($87.03\%$). The uncertainty-based weighting combined with the non-linear Z-space mapping effectively suppresses high-entropy noise channels.

## Source Code

We provide a demo code package in `source_code.zip`. This implementation reproduces the **Asymmetric View Scaling** experiment (corresponding to Figure 3), demonstrating how SAEF maintains robustness while baselines collapse when one view is perturbed by a scale factor $T$. Instructions for running the demo are provided in the `README.md` file.

