# OpenReview forum: "Beyond Magnitude: Scale-Invariant Evidential Fusion for Multi-View Classification"
_ICML.cc/2026/Conference — ICML 2026 regular_

### Official Review · Reviewer_i22T · 2026-03-10

**Soundness:** 3
**Presentation:** 3
**Significance:** 4
**Originality:** 4
**Overall Recommendation:** 5
**Confidence:** 4

**Summary:**

This work identifies a critical limitation in Evidential Multi-view Classification (EMC): the Scale Mismatch Problem. The authors argue that existing methods erroneously conflate evidence magnitude with semantic reliability, rendering them susceptible to semantic hijacking. To address this, the authors propose Scale-Invariant Evidential Fusion (SAEF), a framework that decouples confidence from scale via instance-wise standardization. Experimental results demonstrate that SAEF outperforms baselines in robustness against scale perturbations, conflicting views, and sensor noise.

**Compliance With Llm Reviewing Policy:**

Affirmed.

**Final Justification:**

The author's response addressed my main concerns, so I have decided to raise my rating to Accept.

**Key Questions For Authors:**

1. There is a discrepancy regarding the backbone network. Section 6.1 states that ResNet-18 is used, whereas Appendix C.2 claims ResNet-50 is employed. Which backbone was actually used to generate the results in Table 1 and the figures?

2. How sensitive is SAEF when K is small (e.g., binary classification), it is suggested to provide a deeper analysis.

3. The decoupling of magnitude and confidence might be detrimental in scenarios where low magnitude inherently signifies low data quality (e.g., low-contrast medical images representing ignorance). Does SAEF risk "hallucinating" high confidence for these inherently weak-signal samples by forcing them to a standard scale? A discussion on this potential limitation is suggested.

Overall, the claim that “Magnitude is not equivalent to Reliability” offers valuable insights for the field of evidential multi-view learning. If the authors provide a convincing response to the points raised, I would be willing to upgrade my rating.

**Limitations:**

The authors have discussed the potential negative societal impact in main text. Moreover, the limitations can be found in the Weaknesses part.

**Strengths And Weaknesses:**

Strengths:

1. The paper identifies and theoretically proves the Scale Mismatch Problem in EMC. This finding is novel and offers a new perspective on multi-view fusion within the context of evidential deep learning.

2. The theoretical analysis of the proposed SAEF framework is rigorous, providing solid proofs for both global scale invariance and robustness against asymmetric dominance.

3. The experimental evaluation is comprehensive, covering scale perturbations, sensor noise, and view conflicts. The empirical results align well with the theoretical claims.

Weaknesses:

1. The experiments rely on artificial Temperature Scaling with an extreme range ($T \in [0.01, 100]$). While the theoretical analysis of scale sensitivity is sound, such severe linear shifts in logits are unlikely to occur in real-world applications unless there is catastrophic sensor corruption. The applicability of the method to more natural distribution shifts remains unclear.

2. The proposed standardization depends on calculating instance-wise statistics ($\mu, \sigma$) across $K$ classes. For binary classification tasks like MURA ($K=2$), estimating these statistics from only two data points is inherently unstable. The paper lacks a deep analysis of how this statistical variance affects performance in low-class-count scenarios.

3. Key ablation studies and some derivations of the "evidence-space" forms for DS fusion are relegated to the appendix. Moving the core derivations regarding the quadratic sensitivity of DS fusion and ablation results to the main text would strengthen the paper’s readability.

---

> ### Author Rebuttal · Authors · 2026-03-30
>
> We sincerely thank you for your positive evaluation and for recognizing the novelty of identifying the Scale Mismatch Problem, the rigor of our theoretical proofs, and the comprehensiveness of our experiments. Below, we address your concerns point by point.
>
> **Response to W1**
>
> We appreciate the reviewer's perspective. We want to clarify that the artificial temperature scaling ($T \in [0.01, 100]$)  was specifically designed as a stress test to empirically validate our theoretical boundaries under extreme, catastrophic sensor corruption (as analyzed in Theorems 5.1 and 5.2). However, SAEF’s applicability extends to both natural distribution shifts and the types of catastrophic sensor corruptions you rightly noted:
>
> - As demonstrated in our clean data evaluation (Table 1), we tested SAEF on standard, unperturbed benchmarks without applying any artificial scaling. In highly heterogeneous tasks like the SUNRGBD dataset (RGB vs. Depth), the natural logit scales of completely different physical views are inherently misaligned. SAEF achieves its most pronounced performance gain here, outperforming the runner-up by 2.14% in classification accuracy, precisely because it naturally resolves these inherent scale mismatches without requiring manual calibration.
>
> - Furthermore, regarding the real-world sensor corruptions you mentioned, our Sensor Failure experiment (Fig. 5) explicitly simulates this exact scenario. Rather than merely applying an artificial uniform multiplier, we replaced an entire view with pure, unstructured Gaussian noise to mimic a broken sensor. SAEF successfully recognizes that this natural noise lacks informative structural variance ($\sigma < \epsilon$), mathematically clamping the output to safely filter it out.
>
> Therefore, while the extreme $T$ scaling proves our absolute mathematical limits, our strong performance on both clean heterogeneous data (Table 1) and Gaussian noise (Fig. 5) validates SAEF's practical necessity for real-world applications.
>
> **Response to W2 & Q2**
>
> We deeply appreciate this theoretically insightful question. From a classical statistical estimation perspective, calculating variance from two points is indeed generally unstable. However, within our specific formulation, **the Instance-wise Evidential Standardization (Eq. 7) acts as a deterministic geometric projection rather than a population parameter estimation.**
>
> For binary classification ($K=2$), let the raw logit vector be $z = [z_1, z_2]$. The instance-wise mean is exactly $\mu = (z_1+z_2)/2$, and the standard deviation is exactly $\sigma = \sqrt{\frac{(z_1-\mu)^2 + (z_2-\mu)^2}{2}} = \frac{|z_1-z_2|}{2}$. Provided the view is informative ($\sigma > \epsilon$), substituting these into our Eq. 7 algebraically simplifies the standardized logits to exactly: $\hat{z} = \beta \cdot [\text{sign}(z_1-z_2), \text{sign}(z_2-z_1)]$. This implies that $\hat{z}$ will strictly output either $[+\beta, -\beta]$ or $[-\beta, +\beta]$.
>
> **Therefore, rather than suffering from statistical instability, SAEF degenerates into a stable, strict sign-preserving operator in binary tasks**. It deterministically strips away all erratic magnitude fluctuations and preserves solely the pure semantic decision direction. There is zero variance or instability in this bounded output.
>
> Moreover, as shown in Table 1 and Fig. 3-5, the experimental results on the binary MURA dataset empirically confirm that the $K=2$ scenario introduces no detrimental statistical variance.
>
> **Response to W3**
>
> We appreciate this constructive suggestion. Due to the strict page limits of the initial submission, we unfortunately had to relegate these important details to the appendix. In the revised manuscript, we will optimize our formatting to enhance overall readability.
>
> **Response to Q1**
>
> We apologize for this typographical error. All results in Table 1 and the figures were indeed generated using ResNet-18, and we will correct the misstatement in Appendix C.2 in the revised manuscript.
>
> **Response to Q3**
>
> We appreciate this clinically relevant insight. SAEF strictly avoids "hallucinating" confidence for inherently weak signals (e.g., low-contrast images) via its Variance Gating mechanism ($\max(\sigma, \epsilon)$ in Eq. 7).
>
> Low-quality data inherently lacks structural distinctiveness across classes, meaning its variance approaches zero ($\sigma \to 0$). As proven in Theorem 5.3, when $\sigma$ drops below the noise floor $\epsilon$, our gating mechanism clamps the denominator. Consequently, the standardized logits converge to zero ($\hat{z} \to 0$), forcing the evidential uncertainty to saturate to its maximum limit ($u \to u_{max}$).
>
> Thus, instead of hallucinating confidence, SAEF deterministically outputs "ignorance" and safely filters out the uninformative view ($w \to 0$). This is empirically validated in our Sensor Failure experiment (Fig. 5), where SAEF consistently flags pure noise with high reliability (e.g., $>87$\% Detection AUROC on the AVE dataset).

---

> > ### Author Rebuttal · Reviewer_i22T · 2026-04-01
> >
> > I thank the authors for their detailed responses. All of my concerns have been addressed, and I am happy to raise my score.

---

> > > ### Author Response · Authors · 2026-04-01
> > >
> > > We would like to express our gratitude to the reviewer for the constructive comments throughout the review process. We are very encouraged to know that our responses have addressed all your concerns. Thank you for your time and for the support in raising the evaluation of our work.

---

### Official Review · Reviewer_f8Wp · 2026-03-11

**Soundness:** 4
**Presentation:** 3
**Significance:** 4
**Originality:** 4
**Overall Recommendation:** 6
**Confidence:** 5

**Summary:**

This paper proposes Scale-Invariant Evidential Fusion (SAEF), a framework that explicitly decouples reliability from evidential magnitude, which can theoretically ensure strict scale invariance and noise suppression. Empirical results demonstrate that SAEF exhibits superior stability against severe scale perturbations, sensor noise, and multi-view conflicts.

**Compliance With Llm Reviewing Policy:**

Affirmed.

**Final Justification:**

This paper proposes SAEF, a framework that explicitly decouples reliability from evidential magnitude, thereby theoretically ensuring strict scale invariance and noise suppression. In the rebuttal phase, all my concerns have been addressed, particularly C1, which I initially thought limited the generalizability of the proposed method. I believe this is a solid paper that will contribute to the multi-view deep learning community, especially in trustworthy multi-view classification. Therefore, I recommend a Strong Accept.

**Key Questions For Authors:**

I commend the authors for providing the source code. Upon running the demo on the AVE dataset, I verified the robustness trends. However, I noticed a minor numerical drift at extreme scales: accuracy shifts from 82.8\%$\pm$1.8 ($T \in [0.01, 50]$) to 82.9\%$\pm$1.6 at $T=100$.

Is this deviation caused by the scaled logits hitting the noise floor threshold ($\epsilon=10^{-2}$) as defined in Theorem 5.1, or is it purely a floating-point precision artifact? While statistically negligible, clarifying this would strengthen the theoretical boundary analysis.

**Limitations:**

Yes, the authors have discussed.

**Strengths And Weaknesses:**

S1: In evidential multi-view learning, the estimated reliability is typically tied to the evidence magnitude. The authors identify a critical “Scale Mismatch Problem” and theoretically prove that existing evidential fusion rules are highly sensitive to magnitude drifts. Consequently, the fusion process can be easily hijacked by scale perturbations, particularly by arbitrarily inflated noisy or conflicting views. In my view, identifying and addressing this limitation is both novel and crucial for advancing trusted multi-view classification.

S2: The authors provide rigorous theoretical analyses and extensive empirical evaluations. The experiments cover standard scenarios as well as challenging conditions like severe scale perturbations, sensor noise, and multi-view conflicts, which align perfectly with the core objectives of trusted multi-view learning based on evidential fusion.

Although this paper features a clear motivation, rigorous problem formulation, and well-designed experiments, I have the following concerns:

C1: The Scale Mismatch Problem is well-motivated, but the method used to simulate scale perturbations by applying a uniform scalar multiplier to the entire logit vector of a view seems somewhat idealized. Sensor malfunctions might introduce feature-specific magnitude distortions rather than a simple global scalar shift across all classes. It would be beneficial if the authors could discuss how SAEF handles or generalizes to more complex scale distortions.

C2: The “Related Work” section is relatively brief and heavily focused on Evidential Multi-view Deep Learning. I suggest expanding this section to include a broader discussion of multi-view fusion methods that utilize other uncertainty estimation techniques to provide a more comprehensive context for the proposed SAEF.

C3: The ablation study (e.g., Table 3) could benefit from a deeper analytical discussion in the main text. For instance, it is intriguing that combining instance normalization with Sum/Mean fusion still results in a very low Noise AUROC. A more thorough explanation of why linear aggregation fails to reject standardized noise, and specifically how Stouffer fusion's statistical properties resolve this, would significantly enhance the paper's depth.

---

> ### Author Rebuttal · Authors · 2026-03-30
>
> We deeply appreciate your thorough evaluation, strong support, and especially your dedication to running our source code. Below, we address your concerns point by point.
>
> **Response to C1**
>
> We sincerely thank the reviewer for this insightful comment. We utilized this uniform scalar multiplier ($1/T$) primarily to establish a rigorous mathematical proof for global scale invariance (as shown in Theorem 5.1). **In practice, SAEF is inherently designed to handle and generalize to much more complex, feature-specific scale distortions:**
>
> - SAEF does not rely on global scale assumptions. The Instance-wise Evidential Standardization (Eq. 7) dynamically computes the statistical moments ($\mu^v, \sigma^v$) for each sample across its class dimensions. If a sensor malfunction introduces feature-specific additive distortions, the instance-wise mean centering ($z^v - \mu^v$) effectively neutralizes these shifts. Ultimately, SAEF evaluates the structural consensus (the relative sharpness or statistical significance of the distribution) rather than relying on uncorrupted raw magnitudes.
>
> - Furthermore, we have already empirically evaluated SAEF under complex, non-uniform scale distortions in our robustness analysis (Section 6.4). Specifically:
>
>   - **Sensor Failure (Gaussian Noise, Fig. 5)**: We explicitly simulated complex, random magnitude distortions by replacing an entire view with standard Gaussian noise. Unlike a uniform scalar shift, this distortion introduces random, feature-specific perturbations across all classes. As theoretically proven (Theorem 5.3) and empirically shown, SAEF’s Variance Gating mechanism inherently recognizes that this random noise lacks structural, informative variance. It clamps the output to ensure the complex noise is safely rejected, preventing performance collapse and consistently flagging the anomaly with a high Detection AUROC.
>
>   - **Semantic Hijacking (Fig. 4)**: We also simulated targeted, asymmetric structural distortions by altering the semantic target of a view and injecting extreme magnitudes into specific conflicting classes. SAEF remains highly robust against these feature-specific adversarial distortions, preventing semantic hijacking while baselines suffer from false dominance and catastrophic performance degradation.
>
> We highly appreciate this feedback. We will explicitly add a discussion in the revised manuscript outlining how SAEF's statistical mechanisms effectively handle these complex, real-world sensor malfunctions.
>
> **Response to C2**
>
> We thank the reviewer for this constructive suggestion. In the revised manuscript, we will expand the "Related Work" section to encompass a broader discussion of multi-view fusion methods utilizing alternative uncertainty estimation techniques, thereby providing a more comprehensive context for SAEF.
>
> **Response to C3**
>
> We highly appreciate this insightful observation. The analytical reason why combining instance normalization with linear fusion (Sum/Mean) fails to reject noise (Noise AUROC roughly 27-28%) stems from the fundamental difference between passive mixing and active filtering.
>
> When a view contains pure noise, our standardization mechanism successfully flattens its distribution, pushing its evidential uncertainty to the maximum ($u \to u_{max}$). However, linear aggregation rules (Sum/Mean) operate by passively mixing raw evidence. They blindly accumulate this flattened, uninformative evidence (e.g., the base activation bias $a(0)$) into the joint pool, which passively dilutes the confident predictions of the reliable views and causes the failure in noise detection.
>
> In contrast, Stouffer fusion utilizes active filtering. It converts evidential probabilities to the canonical Gaussian domain (Z-space) and dynamically weights them by view-specific confidence ($w = 1 - u$). For the standardized noisy view, $u \to u_{max}$ inherently forces its fusion weight to zero ($w \to 0$). As rigorously proven in Theorem 5.3 (Eq. 33), this statistical property allows Stouffer fusion to algebraically eliminate the noise term ($\lim_{w_{noise} \to 0} Z^{joint} = Z^{rel}$) rather than just diluting it.
>
> We fully agree that this analytical depth significantly enhances the paper. Following your suggestion, we will explicitly incorporate this detailed mathematical explanation comparing linear aggregation and Stouffer fusion into the main text.
>
> **Response to Q1**
>
> You are absolutely correct. The minor drift is not a precision artifact but the activation of the noise floor ($\epsilon = 10^{-2}$) defined in Theorem 5.1. At $T=100$, the severely compressed variance for a few boundary samples drops below $10^{-2}$, leading to $\frac{1}{T}\sigma(z^v) < \epsilon$. This violates the theorem's precondition and triggers the Variance Gating mechanism. To empirically verify your point, we lowered the threshold to $\epsilon = 10^{-3}$ in the demo. As expected, the accuracy remained strictly constant at 82.8% with absolutely zero drift.

---

> > ### Author Rebuttal · Reviewer_f8Wp · 2026-04-03
> >
> > I thank the authors for their responses. All my concerns have been addressed, particularly C1, which I initially thought limited the generalizability of the proposed method. I believe this is a solid paper that will contribute to the multi-view deep learning community, especially in trustworthy multi-view classification. Therefore, I will raise my score.

---

> > > ### Author Response · Authors · 2026-04-03
> > >
> > > We sincerely thank you for your time and effort in the rebuttal process. We are delighted that our responses, especially the explanations regarding the generalizability of our method (C1), resolved your initial concerns. Your constructive feedback has been invaluable in improving the quality of our paper. We truly appreciate your positive evaluation and your strong support for our contribution to the multi-view learning community!

---

### Official Review · Reviewer_Yd1e · 2026-03-11

**Soundness:** 2
**Presentation:** 3
**Significance:** 2
**Originality:** 2
**Overall Recommendation:** 4
**Confidence:** 3

**Summary:**

This paper studies the problem of scale inconsistency in multi-view classification, where different views produce logits with significantly different magnitudes. The authors argue that such scale mismatches lead to unstable or biased fusion results when conventional aggregation strategies are applied. In particular, views with larger logit magnitudes may dominate the final prediction, regardless of their reliability. To address this issue, the paper proposes a Scale-Invariant Evidential Fusion framework. The method first transforms logits from each view into evidential representations and then performs statistical normalization to remove scale discrepancies across views. The normalized evidence is subsequently aggregated through an evidential fusion mechanism that estimates class probabilities and uncertainty using a Dirichlet formulation. The results demonstrate that SAEF consistently improves classification performance compared with existing multi-view fusion methods, especially in scenarios where view-specific scales differ significantly.

**Compliance With Llm Reviewing Policy:**

Affirmed.

**Key Questions For Authors:**

Overall, I appreciate this paper. My main concerns and questions focus on the following:

1. How does the proposed SAEF method behave when the magnitude difference between views becomes extremely large?

2. What is the computational overhead of the proposed normalization and evidential fusion steps compared with standard fusion strategies (e.g., logit averaging)?

**Limitations:**

Please see the weaknesses.

**Strengths And Weaknesses:**

Strengths

1. The proposed approach is grounded in evidential deep learning, which provides a principled probabilistic framework for modeling prediction uncertainty. The use of Dirichlet distributions to represent evidence is well established in prior work, and the authors extend this formulation to multi-view fusion in a reasonable way.

2. The method description is organized logically, guiding the reader from the issue of logit magnitude imbalance to the proposed normalization and evidential aggregation framework. The related work section also provides sufficient context for the study and positions the paper within the literature on multi-view learning and evidential deep learning.

3. By highlighting the scale mismatch issue in multi-view logits, the paper draws attention to a problem that is often overlooked in standard fusion strategies. The proposed scale-invariant evidential fusion approach offers a potentially useful solution that could be applied in other multi-view or multimodal systems.

Weaknesses

1. Although the paper provides intuition about why scale normalization improves fusion robustness, the theoretical analysis remains relatively high-level. A more rigorous analysis of the statistical properties of the normalization step or its effect on evidential uncertainty would strengthen the technical contribution.

2. The experiments mainly demonstrate improved classification accuracy, but the paper could provide more detailed analysis on: 1）robustness under extreme scale imbalance. 2) behavior when one view is highly noisy or adversarial 3) uncertainty calibration quality. Such analyses would help better understand the robustness advantages of the proposed method.

3. The paper does not clearly discuss the computational overhead introduced by the normalization and evidential fusion process. Although the method appears lightweight, as the authors discussed in line 272, a brief runtime comparison would improve the practical evaluation.

---

> ### Author Rebuttal · Authors · 2026-03-30
>
> We sincerely thank you for the constructive feedback and for recognizing the value of addressing the often-overlooked scale mismatch problem within the principled framework of EMC. Below, we address your concerns regarding theoretical rigor, extreme robustness experiments, and computational overhead.
>
> **Response to W1**
>
> We thank you for highlighting the need for a more concentrated theoretical exposition. We would like to clarify that the analysis regarding the statistical properties of the normalization step and its exact mathematical effect on evidential uncertainty ($u$) is formalized in our theorems and Appendix, though we agree it can be highlighted more prominently. Specifically, the normalization step’s effect on uncertainty is rigorously bounded by two analytical properties in our framework:
>
> - **Invariance Property (Effect on $u$ under scaling)**: As formally proven in Theorem 5.1 and Appendix B.2, the normalization step introduces a strict mathematical decoupling between the arbitrary input scale $T$ and the output uncertainty. Because the standardization cancels out $1/T$ (Eq. 28), it is strictly guaranteed that $\hat{u}^v(T) \equiv \hat{u}^v(1)$, eliminating the "Evidential Scale Ambiguity".
>
> - **Asymptotic Bounding Property (Effect on $u$ under noise):** The statistical properties of the Variance Gating mechanism (controlled by the noise floor $\epsilon$) can be found in the proof of Theorem 5.3 (Appendix B.2). We show that when a view lacks statistical informativeness ($\sigma^v < \epsilon$), the normalization step algebraically collapses the logits $\hat{z}^v \to 0$ (Eq. 31). We mathematically prove that this property explicitly forces the evidential uncertainty to saturate to its maximum limit ($u_{noise} \to u_{max}$), thereby flagging the uninformative view and enhancing the fusion performance.
>
> We completely agree that integrating these statistical properties directly into the main text will strengthen the technical contribution, and we will add the above analysis to the revised manuscript.
>
> **Response to W2 & Q1**
>
> We appreciate your insight and completely agree with the importance of these scenarios. In our robustness evaluations (Section 6.4, Figures 3-5), we specifically simulated these extreme and adversarial conditions:
>
> - **Extreme Scale Imbalance (Fig.3)**: Our experiments evaluate a scaling factor spanning $T \in [0.01,100]$, which tests a massive dynamic range of four orders of magnitude. At the extreme end ($T=0.01$), the scaled view is subjected to a 100-fold amplification relative to the unperturbed views. As shown in Fig.3, such extreme magnitude differences cause baseline methods to suffer a catastrophic "Inverted-V" accuracy collapse. In contrast, SAEF remains structurally invariant and perfectly stable.
>
>     Beyond this empirical 100-fold test, Theorem 5.2 mathematically guarantees SAEF's strict invariance even as the magnitude difference approaches infinity ($\lim_{T \to 0}$). Moreover, as $T \to \infty$, the suppressed feature variance approaches zero. As proven in Theorem 5.3, SAEF's Variance Gating mechanism forces the standardized logits to collapse ($\hat{z} \to 0$) if $\sigma^v < \epsilon$, saturating the uncertainty ($u \to u_{max}$). Consequently, the fusion weight of the suppressed view asymptotically approaches zero ($w \to 0$), safely filtering it out without dragging down the joint prediction.
>
> - **Highly Noisy/Adversarial Views (Fig.4&5)**: In Section 6.4, we simulated "Semantic Hijacking" (an adversarial view with conflicting labels and a 100x amplified magnitude) and "Sensor Failure" (pure Gaussian noise with amplified magnitude). While baselines are hijacked (Accuracy $\approx 0\%$), SAEF clamps the noise and correctly outputs high evidential uncertainty, yielding a Detection AUROC of over 93% for the conflicting view and over 87% for the highly noisy view (as detailed in Tables 3-5). **This high Detection AUROC under severe perturbation directly demonstrates our model's superior uncertainty calibration quality.**
>
> **Response to W3 & Q2**
>
> We thank you for pointing this out. While we discussed the theoretical asymptotic complexity O(VK) in the Remark at the end of Section 5, providing empirical runtime is indeed practically valuable.
>
> Here we conducted a runtime test on a single NVIDIA RTX 4090 GPU. The table below reports the average inference time per sample (in milliseconds) across the 5 folds:
> | Method | AVE | CheXpert | SUNRGBD | MURA |
> | :--- | :---: | :---: | :---: | :---: |
> | **Ours** | 0.45 | 2.54 | 4.94 | 5.45 |
> | **DS** | 0.42 | 2.60 | 4.83 | 5.43 |
> | **Sum** | 0.49 | 2.53 | 4.98 | 5.44 |
> | **Mean** | 0.39 | 2.58 | 4.95 | 5.32 |
> | **Weighted** | 0.41 | 2.56 | 4.83 | 5.48 |
>
> As empirically shown, the computational overhead introduced by SAEF is practically negligible (e.g., a difference of ~0.1ms compared to the logit Mean). We will include this runtime comparison table in the revised Appendix to explicitly demonstrate its lightweight nature.

---

> > ### Author Rebuttal · Reviewer_Yd1e · 2026-04-03
> >
> > Thank you for the rebuttal. My concerns have been addressed, and thus I will retain my original score.

---

> > > ### Author Response · Authors · 2026-04-03
> > >
> > > We sincerely thank you for engaging in the rebuttal process and for confirming that your concerns have been adequately resolved. Your insightful feedback and suggestions have been invaluable in refining our manuscript. Thank you again for your time, effort, and continued support of our work.

---

### Official Review · Reviewer_nSEo · 2026-03-13

**Soundness:** 4
**Presentation:** 3
**Significance:** 3
**Originality:** 3
**Overall Recommendation:** 5
**Confidence:** 2

**Summary:**

This paper proposes a new fusion operator in evidential deep learning.
The authors show that existing fusion operators can be fooled by simply scaling classifier logit magnitudes, resulting in outputs that may be overconfident or underconfident.
To address this issue, the authors propose a scale-invariant fusion operator in which the logits are normalized before fusion.
Desirable theoretical properties are provided for this new fusion operator.
Experimental results show that this scale-invariant operator outperforms existing ones, particularly on challenging benchmarks.

**Compliance With Llm Reviewing Policy:**

Affirmed.

**Final Justification:**

This seems like a solid paper.

**Key Questions For Authors:**

Q1. I don't fully understand the experimental setups and implications; let's begin with Table 1. Based on the dataset descriptions in Appendix C1, for each dataset, are you proposing to train a separate model for each view, and then combine (fuse) their predictions?

Q2. If my above interpretation is correct, I can understand accuracy, but what does AUROC mean in this context?

Q3. It is surprising and impressive to me that SAEF is so dominant over existing methods on all datasets. Can you give me some intuition as to why this might be the case?

**Strengths And Weaknesses:**

## Strengths

S1. A key weakness in the existing literature is identified.

S2. The proposed solution is theoretically well-motivated.


## Weaknesses

W1. This paper is broadly well executed, but there are a few questions I would like clarification on; see below.

---

> ### Author Rebuttal · Authors · 2026-03-30
>
> We sincerely thank you for your encouraging feedback and deeply appreciate your strong support. Below, we provide detailed clarifications to your excellent questions.
>
> **Response to Q1**
>
> We thank you for this accurate observation. Conceptually, your interpretation is correct. Although we train all views concurrently within a unified network architecture, each view is processed by an independent, non-weight-sharing branch. More importantly, our total objective (Eq.13) strictly computes and minimizes view-specific losses independently, without relying on a joint fusion loss.
>
> In standard fusion-driven optimization (where loss is computed on the fused output), a fast-learning view can mathematically suppress the gradients of weaker views, artificially shrinking their evidence magnitudes. By employing Eq.13, each branch fully exploits its view concurrently, preserving its natural but inherently unaligned evidential scale. Consequently, robustly aggregating these uncalibrated evidences necessitates a scale-invariant mechanism like SAEF, avoiding forced artificial model-wide alignment.
>
> **Response to Q2**
>
> We sincerely thank you for pointing this out. We realize using the term "AUROC" across sections caused confusion, as it evaluates two distinct capabilities in our experiments:
>
> - **Classification AUROC (Table 1):** This measures standard multi-class classification performance. Computed using fused predicted probabilities against ground-truth labels on clean test data, it evaluates the ranking quality of the predicted probability distribution. As you noted, this metric is fully compatible with any fusion paradigm. (Note: We apologize for a typo in Section 6.2 where we mistakenly referred to this specific metric as "failure detection capabilities", which we will correct in the revision).
>
> - **Detection AUROC (Fig.4 & 5, Tables 3-5):** Used in our robustness analysis, this measures the model's Conflict/Noise Detection capability, evaluating if the model *knows when it does not know*. We treat clean samples as the negative class and the synthetically corrupted ones (e.g., Semantic Hijacking (Fig.4) or Sensor Failure (Fig.5)) as the positive class, using evidential uncertainty (vacuity, $u$) as the score. A high Detection AUROC means the model outputs low uncertainty for clean inputs but reliably triggers high uncertainty when subjected to conflict or noisy samples, effectively avoiding the "False Confidence" trap.
>
> By explicitly differentiating these two metrics, we demonstrate that SAEF not only improves predictive accuracy (Table 1) but also guarantees system safety by reliably flagging conflicts or noise (Fig.4 & 5). We will explicitly distinguish between "Classification AUROC" and "Detection AUROC" in the revised manuscript to prevent further confusion.
>
> **Response to Q3**
>
> We appreciate your question. The intuition behind SAEF's consistent advantage is rooted in how it resolves the "Scale Mismatch Problem". In standard paradigms, fusion is explicitly dictated by absolute view magnitudes. This causes the model to blindly trust amplified views, leading to incorrect uncertainty estimation (False Confidence), catastrophic accuracy collapse, and a total failure to reflect underlying conflicts or noise. SAEF overcomes this through the following mechanisms:
>
> - **Theoretical Intuition**: As proved in Proposition 3.2 and Corollary 3.3, existing methods are sensitive to scale imbalances; a noisy view with an arbitrarily large magnitude will mathematically hijack the fusion. SAEF’s dominance comes from abandoning this magnitude-dependent paradigm. As proved in Theorem 5.2, SAEF mathematically guarantees that the fused probability distribution remains anchored to the true consensus of the views, remaining strictly invariant even if a conflicting view is infinitely amplified.
>
> - **Empirical Intuition on Clean Data**: In highly heterogeneous multi-view tasks, the natural logit scales of different views often vary significantly. For instance, on the SUNRGBD dataset (RGB vs. Depth), scale discrepancies are inherently severe. SAEF’s dominance is most pronounced here (outperforming the runner-up by 2.14% in Table 1) precisely because it operates in a scale-invariant statistical manifold. It prevents the mathematically "louder" view from blindly suppressing the weaker one, thereby safely unlocking the genuine complementary information across all views.
>
> - **Empirical Intuition under Perturbation**: When a corrupted view is artificially amplified ($T<1$, Fig.4 & 5), baselines suffer catastrophic accuracy and Detection AUROC collapse (≈0%). They blindly trust "loud" anomalies, generating inflated evidence $S$ and artificially low uncertainty ($u=K/S$). Conversely, SAEF’s variance gating recognizes that amplified noise lacks structural variance. It mathematically clamps these uninformative signals to correctly output high uncertainty, ensuring a stable ACC curve and reliably high Detection AUROC under extreme scale perturbations.

---

> > ### Author Rebuttal · Reviewer_nSEo · 2026-04-03
> >
> > Thank you for the comprehensive response. I will maintain my "Accept" score. Good luck with this submission.

---

> > > ### Author Response · Authors · 2026-04-04
> > >
> > > We sincerely thank you for taking the time to review our response. We are very glad that our response addressed your concerns. We deeply appreciate your continued support, positive evaluation, and well wishes for our submission! Thank you again!

---

### Decision · Program_Chairs · 2026-04-30

**Decision:**

Accept (regular)

**Comment:**

This paper received ratings of 1 Strong Accept, 1 Accept, and 1 Weak Accept. Following the rebuttal, reviewers indicated that their concerns were fully resolved, leading to a unanimous positive consensus. The reviewers found the paper well-motivated with a clear problem formulation and comprehensive experiments, highlighting the novelty of the scale-invariant evidential fusion mechanism. Given the full resolution of concerns and the resulting consensus, the AC has decided to accept this paper.